# PGP-14 establishes a polar lipid permeability barrier within the *C. elegans* pharyngeal cuticle

**Muntasir Kamal[1,2], Levon Tokmakjian[2,3], Jessica Knox[1,2], Duhyun Han[2,3], Houtan Moshiri[1,2], Lilia Magomedova[4], Ken CQ Nguyen[5], Hong Zheng[1,2], Andrew R. Burns [1,2], Brittany Cooke[1,2], Jessica Lacoste[1,2], May Yeo[1,2], David H. Hall [5], Carolyn L. Cummins[4], Peter J. Roy [1,2,3]***

1 Department of Molecular Genetics, University of Toronto, Toronto, Ontario, Canada, 2 The Donnelly Centre for Cellular and Biomolecular Research, University of Toronto, Toronto, Ontario, Canada, 3 Department of Pharmacology and Toxicology, University of Toronto, Toronto, Ontario, Canada, 4 Department of Pharmaceutical Sciences, Leslie Dan Faculty of Pharmacy, University of Toronto, Toronto, Ontario, Canada, 5 Department of Neuroscience, Albert Einstein College of Medicine, New York, New York, United States of America

* peter.roy@utoronto.ca

**Data Availability Statement:** All data generated or analyzed during this study are included in this article and its supplementary information files.

## Abstract

The cuticles of ecdysozoan animals are barriers to material loss and xenobiotic insult. Key to this barrier is lipid content, the establishment of which is poorly understood. Here, we show that the p-glycoprotein PGP-14 functions coincidently with the sphingomyelin synthase SMS-5 to establish a polar lipid barrier within the pharyngeal cuticle of the nematode *C. elegans*. We show that PGP-14 and SMS-5 are coincidentally expressed in the epithelium that surrounds the anterior pharyngeal cuticle where PGP-14 localizes to the apical membrane. *pgp-14* and *sms-5* also peak in expression at the time of new cuticle synthesis. Loss of PGP-14 and SMS-5 dramatically reduces pharyngeal cuticle staining by Nile Red, a key marker of polar lipids, and coincidently alters the nematode's response to a wide-range of xenobiotics. We infer that PGP-14 exports polar lipids into the developing pharyngeal cuticle in an SMS-5-dependent manner to safeguard the nematode from environmental insult.

## Author summary

An infinite number of small molecules have the potential to threaten life. Not surprisingly then, animals have evolved multiple mechanisms to defend against such threats. One defense mechanism employed by many animals is the creation of an outer protective layer called the cuticle. Lipids within the cuticle act as a barrier to retard small molecule passage into the underlying tissues. Water-loving small molecules cannot traverse a lipid barrier and fat-loving molecules can get trapped within the barrier. How this lipid barrier is established is incompletely understood. Here, we describe our discovery of a conserved protein called PGP-14 that is expressed in the tissue that makes the cuticle that protects the mouth and pharynx of the nematode worm *C. elegans*. We show that PGP-14 peaks in expression at the end of new cuticle synthesis and is necessary for lipid deposition within it. Without

**Funding:** This work is supported by CIHR grants (153024 and 173448) and a Canada Research Chair (Tier 1) to PJR, an NSERC (RGPIN 2020-07212) to CLC, and an NIH grant (OD 010943) to DHH. MK, LT, DH, HM, HZ, ARB, BC, JL, and MY were supported by PJR's CIHR grants; JK and LM were supported by a Natural Sciences and Engineering Research Council of Canada (NSERC) Alexander Graham Bell Canada Graduate Scholarship. KCQN is supported by DHH's NIH grant. The funders had no role in study design, data collection and analysis, decision to publish, or preparation of the manuscript.

**Competing interests:** The authors have declared that no competing interests exist.

PGP-14, many small molecules adversely accumulate within the animal and consequently kill it. Hence, PGP-14 is a key component employed by the animal to help protect it from small molecule threats.

## Introduction

The cuticles of the roughly 4.5 million ecdysozoan species (animals with exoskeletons) play multiple essential roles. These include serving as substrate to anchor muscles that generate force, providing protection from physical injury, and acting as a permeability barrier at the molecular level. Key to the permeability barrier is lipid. For example, the outer epicuticle layer in the nematode exocuticle is rich in lipid [1–3] and plays important roles in both xenobiotic and pathogen defense [4–7]. Similarly, the inner lipid-rich layer of the nematode eggshell also functions as a xenobiotic permeability barrier [8–10]. How permeability barriers are established is not well understood.

Here, we describe the role of an ABCB-class transporter in establishing the lipid barrier within the pharyngeal cuticle of the nematode *Caenorhabditis elegans*. The pharyngeal cuticle, which we defined as the cuticles of both the buccal cavity and pharynx, contains a chitin-chitosan matrix that likely provides tensile strength to maintain luminal integrity [11–14]. We and others have shown that the cuticle is also highly enriched with intrinsically-disordered proteins [15,16]. At the end of each larval stage, *C. elegans* undergoes a molt whereby all cuticles are shed. As the old cuticle is being shed, a new cuticle is built underneath, and the next developmental stage ensues [17,18]. We have previously established a spatiotemporal blueprint of gene expression that predicts how the pharyngeal cuticle is constructed [16].

Screens performed in our lab and in the labs of others have shown that select small molecules visibly accumulate in the *C. elegans* pharynx [19–21]. We have demonstrated that these molecules form birefringent crystals or globular spheres in association with the anterior pharynx and nowhere else in the animal [20]. We also showed that the crystals are not simply the result of animals ingesting drug precipitate that may be in the media [20]. The growing small molecule crystals perforate the plasma membrane of the adjacent pharynx epithelium and can grow to a size that likely disrupts the ability of smaller larvae to feed [20]. Crystal formation is tightly correlated with larval death [20].

To better understand the factors that facilitate small molecule crystal formation, we previously performed a forward genetic screen for animals that resist the lethality induced by the crystal-forming molecule wact-190 (worm-active molecule #190) [20]. From this screen, we identified three genes that encode components of a presumptive sphingomyelin synthesis pathway, including the terminal enzyme sphingomyelin synthase 5 (SMS-5) [20]. Sphingomyelin is one of four abundant phospholipid classes within the metazoan plasma membrane. Sphingomyelin and phosphatidylcholine reside abundantly within the outer leaflet of the plasma membrane of animals, while phosphatidylserine and phosphatidylethanolamine reside abundantly within the inner leaflet [22]. SMS-5 is expressed in the pharyngeal epithelial cells that likely generate the anterior pharyngeal cuticle and is necessary for abundant sphingomyelin production by pharynx cells [16,20].

Here, we report on a fourth complementation group called *pgp-14* that we found to resist crystal formation. *pgp-14* encodes a putative p-glycoprotein that is homologous to human ABCB transporters that include ABCB4 (otherwise known as MDR3) [23]. ABCB4 exports phosphatidylcholine (PC) lipids from hepatocytes into the adjacent lumen called bile canaliculi

[24]. There, the lipids combine with bile acids to form micelles, which aid in the digestion of fats and other hydrophobic nutrients [25].

We find that *pgp-14* is expressed exclusively from the epithelial cells that are adjacent to the anterior pharyngeal cuticle. We also find that *pgp-14* and *sms-5* are co-expressed in pharynx cells and that both are upregulated simultaneously during new cuticle synthesis. Mutations in both genes coincidently sensitize the animals to over 100 distinct bioactive compounds. Upon staining animals with a variety of dyes that bind the pharyngeal cuticle, we find that both *pgp-14* and *sms-5* mutants have a specific deficit of Nile Red staining within the pharyngeal cuticle relative to the wild type. Nile Red fluoresces red only in the presence of polar lipids [26]. We infer that PGP-14 and SMS-5 are necessary for the establishment of a polar lipid barrier within the pharyngeal cuticle.

Our data suggests that PGP-14 pumps polar lipids into the developing cuticle in a manner that is dependent on a plasma membrane rich in sphingomyelin. We posit that it is this lipid barrier that acts as a sink for the hydrophobic small molecules that form the birefringent crystals within the pharynx cuticle whilst also acting as a barrier for the accumulation of other small molecules into the animal. Transporting phospholipids into the digestive tract of animals by p-glycoproteins may therefore be an ancient role that is shared by PGP-14 and ABCB4 alike.

## Results

### Small molecule crystals emanate from the non-luminal face of the pharyngeal cuticle

Using simple light microscopy, birefringent crystals can readily be seen in association with the pharynx cuticle of wact-190-treated worms (Fig 1A–1F'). To better understand the locale of crystal formation, we examined wact-190 objects by transmission electron microscopy (TEM). Through the examination of serial sections, we found that unusual objects form in close association with the non-luminal face of the pharyngeal cuticle (Fig 2A–2K). We presume that these unusual objects are the crystals that we see by light microscopy, which also appear to extend from the non-luminal face of the cuticle (see Fig 1F for example). As the objects extend in space, they can be seen either piercing the plasma membrane (Fig 2C, 2C', 2I and 2J) or adjacent to the plasma membrane like a finger pushing on the skin of a balloon (Fig 2F and 2F'). In either case, the extension of the object from the pharyngeal cuticle is coincident with the distortion of the plasma membrane inwards towards the cytoplasm.

The larval stages of *C. elegans* are defined by cuticle molts where the animal sheds its old cuticle and grows a new one underneath [17]. If our interpretation that objects form in association with the pharyngeal cuticle is correct, then removing larval worms from a solution of the object-forming compound should result in the shedding of the crystals along with the old cuticle during the subsequent molt. If instead the crystals initially form within the plasma membrane or in the cytoplasm, then the crystals should remain after the molt. We used a pulse-chase protocol that we recently established [16] to determine whether the crystals are likely associated with the pharyngeal cuticle (see methods for details). The workhorse molecule that we used for most of our crystal studies is the non-fluorescent wact-190 because of its long-term stability. However, for some experiments, including this pulse chase experiment, we used wact-469 because its fluorescent properties make it easier to track. Briefly, we incubated third larval stage (L3) animals with a pulse of the fluorescent crystalizing molecule wact-469 for three hours, then transferred the animals to a rinse plate for 1 hour, and then a chase plate without wact-469 for another 18 hours (Fig 2L). We counted the number of animals with and without fluorescent signal at the beginning and then at the end of the chase period. After the chase, very few animals that were initially L3s had fluorescent signal (Fig 2M–2O). By contrast,

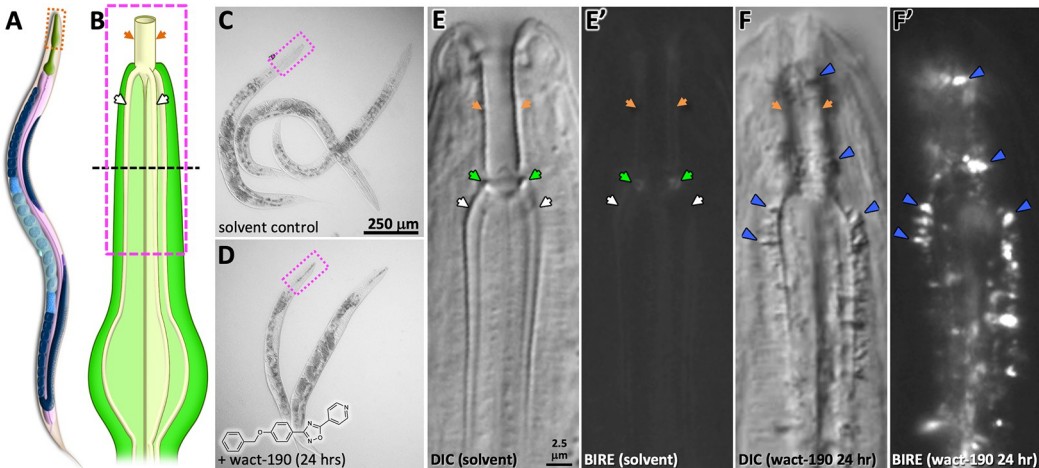

**Fig 1. Small Molecules Crystalize in Association with the Pharynx Cuticle. A.** A schematic of *C. elegans* anatomy (from WormAtlas; used with permission). The pharynx is green. **B.** A schematic of the corpus that is the subject of panels C-F. The central lumen is indicated in dark grey, the channels and buccal cavity are in cream. The black dashed line corresponds to the approximate cross sections shown in Fig 2. **C-D.** Photos of worms taken at the working dissection microscope illustrating the obviousness of wact-190 crystal formation relative to the control. The pink boxed area corresponds to the area highlighted in A and B. The scale bar in C also applies to D. The wact-190 small molecule is shown in D. **E.** Anterior-most region of a worm treated with solvent only. Orange arrows indicate the buccal cavity cuticle, the green arrows highlight the buccal collar, and the white arrows indicate the cuticle that lines the channels. Differential interference contrast (DIC) and birefringent (BIRE) images are shown. The scale bar in E applies to all images in the right-hand side. **F.** Anterior-most region of a worm treated with 30 μM wact-190 for 24 hours. The notation is the same as that in E. Blue arrowheads highlight the birefringent crystals.

when the pulse-chase experiment was done with young adults, which do not molt, the fluorescent signal persisted (Fig 2P and 2R).

We investigated the site of crystal formation using a third approach. Previous work showed that RNAi targeting *mlt-9* results in animals that remain attached to their shed anterior exocuticle and pharyngeal cuticle [16,27]. We investigated whether attached shed pharyngeal cuticle would accumulate fluorescent crystals when incubated in wact-469 for 3 hours and found that it did (Fig 2S and 2T). Importantly, we note that if the crystallizing molecules simply formed in the space between the cuticle and the underlying epithelium, we would not see attachment of the crystals to the partially shed cuticle. Together, these data indicate that wact-190 birefringent crystals form in tight association with the non-luminal face of the pharyngeal cuticle and can distort the plasma membrane of cells that line the pharynx lumen (Fig 2U–2X).

## PGP-14 is needed for small molecule crystal formation

To identify the biological components that facilitate crystal formation, we previously carried out a forward genetic screen of 1.3 million randomly mutagenized wild type *C. elegans* F2 genomes for mutants that resist the lethal effects of wact-190 as previously described [20]. Whole-genome sequencing of the wact-190-resistant strains revealed 29 alleles of *pgp-14* (Figs 3A, 3B and S1). At least 12 of the *pgp-14* alleles are obvious strong reduction-of-function alleles or nulls based on the nature of the mutation (Fig 3). We also found two additional *pgp-14* alleles from the million mutation project [28] and the independently derived deletion allele *pgp-14 (ok2660)* from the knock out consortium [29], each of which confers resistance to wact-190-induced death (Fig 3). We henceforth use the *ok2660* allele as our reference allele and simply refer to it as *pgp-14(0)*.

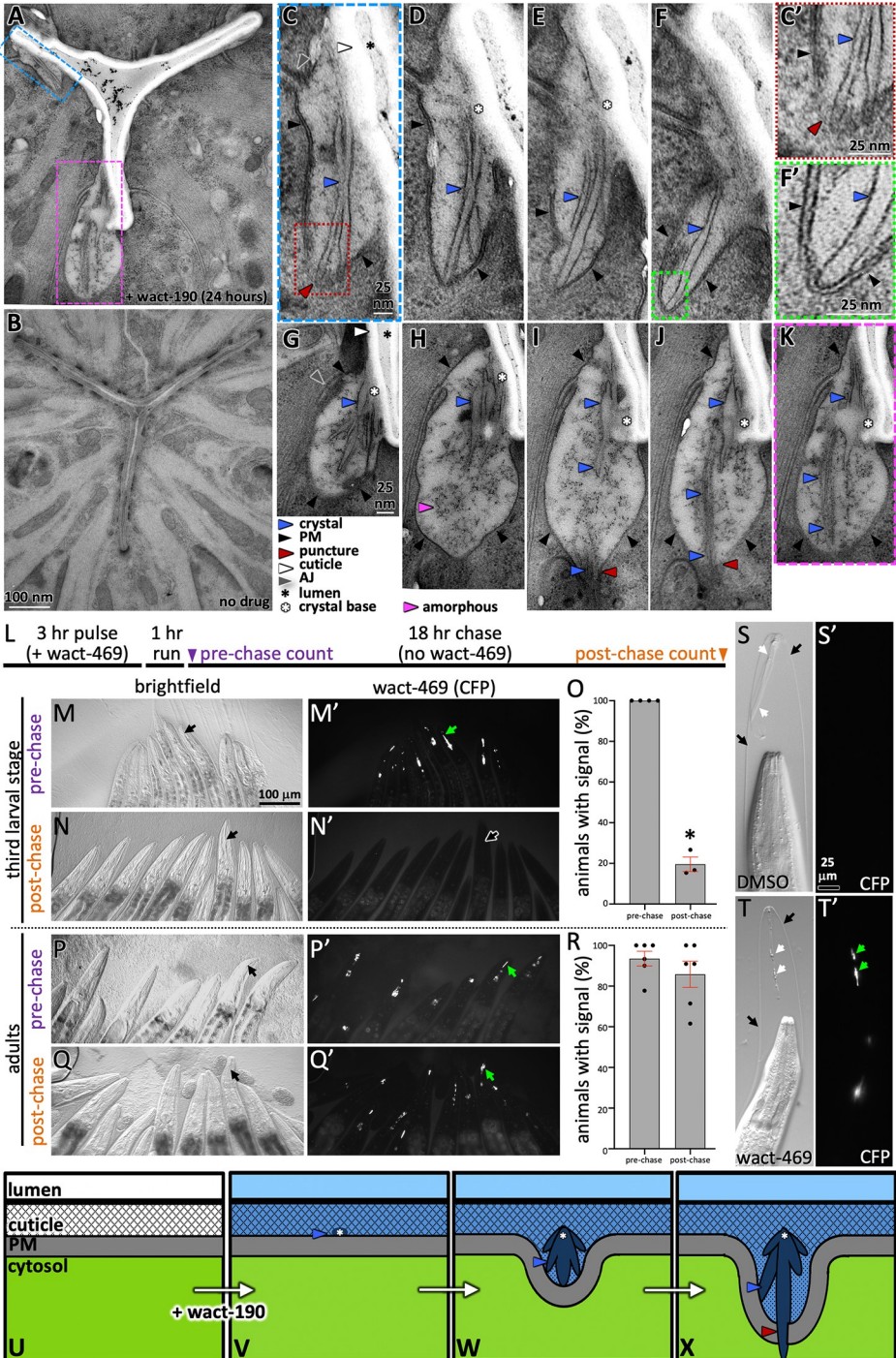

**Fig 2. Wact-190 Crystals Form in Association with the Pharyngeal Cuticle. A-B.** Transmission electron microscopy of cross sections of wild type worms incubated with either 30 μM wact-190 (A) or solvent control (DMSO; B). The scale bar in B also applies to A. **C-F.** Serial sections of the area indicated in the blue box in 'A' ('C' is the actual section of the series highlighted in 'A'). The scale in C applies to C-F. **C'.** An enlargement of the area boxed in 'C' showing how the object (blue arrowhead) is disrupting the plasma membrane. **F'.** An enlargement of the area boxed in 'F' showing how the object (blue arrowhead) is juxtaposed to the plasma membrane (black arrowheads). **G-K.** Serial sections of the area indicated in the pink box in 'A' ('K' is the section of the series highlighted in 'A'). The scale in G applies to G-K. **L.** A timeline of the pulse-chase experiment. **M-N.** Examples of third larval stage animals that have been incubated in fluorescent wact-469 (60 μM) before the chase (M) or after the chase (N). A Cyan Flourescent Protein (CFP) filter was used to visualize wact-469. The scale in M applies to M-Q'. **O.** Quantification of the fraction of animals with

fluorescent wact-469 crystals. The experiment with L3s had two independent repeats, each with two technical repeats, and each with at least nine worms. The experiment with adults had three independent repeats, each with two technical repeats, and each with at least nine worms. The asterisk indicates a significant difference ($p = 1.3E{-}06$) relative to pre-chase using a Student's t-test. Standard error of the mean is shown. **P-R.** Analogous to M-O, but done with young adult animals. **S&T.** *mlt-9(RNAi)* animals with incompletely separated pharyngeal cuticle (black arrows) exposed to either DMSO (S) or wact-469 (60 µM) (T) for 3 hours. wact-469 crystals are indicated with green arrows. The scale in S applies to S-T'. **U-X.** The interpretation of how and where crystals form in relation to the pharyngeal cuticle and the plasma membrane of the underlying marginal cells. The relative concentration of wact-190 is depicted in shades of blue; the darker the shade, the greater the relative concentration. The legend for the arrows and asterisks in U-X are indicated below panels G and K.

In addition to resisting lethality, *pgp-14(0)* mutants resist the crystal formation that is induced by all concentrations of wact-190 tested (Fig 3A). *pgp-14(0)*'s resistance to wact-190's effects can be fully rescued by a transgenic genomic copy of *pgp-14* (Fig 3C–3E) and by expressing YFP-tagged PGP-14 from the *pgp-14* promoter (Fig 3D). This further confirms that wact-190-resistance is conferred by *pgp-14* loss-of-function and not through gain-of-function mechanisms that commonly occur with PGP mutations that confer multi-drug-resistance in yeast and human cells [30,31].

The *C. elegans* genome encodes 60 ABC transporters, 14 of which are predicted to be P-glycoproteins of the ABCB type with 12 transmembrane domains [23] (S2 Fig). We investigated loss-of-function mutants of the 14 other full-length PGPs as well as a variety of other ABC transporters and found that no other ABC-encoding mutant genes aside from *pgp-14* confer robust resistance wact-190 lethality (S3 Fig). We cannot exclude the possibility that if expressed at the correct time in the correct tissues and at high enough levels, that other *C. elegans* PGPs might be able to substitute for PGP-14.

PGP-14 is homologous to the four human ABCBs (S2 Fig), including the multi-drug transporter ABCB1 (otherwise known as MDR1 or PGP) [32,33]. We therefore considered the possibility that PGP-14 may pump out wact-190 and other crystalizing molecules from the pharynx epithelium that underlies the pharyngeal cuticle. If this were true, then we might expect an increase in the abundance of the respective small molecules in tissue of *pgp-14(0)* mutants relative to wild type animals because the crystalizing molecule cannot be pumped out. We tested this hypothesis in multiple ways. First, we measured the accumulation of wact-190 by mass-spectrometry in *pgp-14(0)* mutants and the wild type. There is a significant decrease in wact-190 abundance in the *pgp-14(0)* background relative to wild type, and this deficit is rescued by transgenically-expressing PGP-14 in the background of the *pgp-14(0)* mutant (Fig 3F). Second, we asked whether the fluorescence of object-forming wact-40, wact-43, and wact-469 accumulates in the epithelium of *pgp-14(0)* mutants relative to wild type. We observe no obvious wact-469 accumulation in the tissue surrounding the pharyngeal cuticle in the *pgp-14(0)* mutant background, even when the photos are over-exposed (Fig 3G). Third, our chemoinformatic analysis of the crystal-forming compounds [20] shows that only 8% (4/48) are predicted to be ABCB1/MDR1 substrates (using the Swiss-ADME online tool [34]) (S1 Data). Coincidentally, the fluorescent wact-40 and wact-469 are two of the four crystalizing molecules predicted to be MDR1 substrates, but even these molecules do not accumulate intracellularly in the *pgp-14(0)* mutant (Fig 3G). Together, these observations argue against the idea that PGP-14 is a drug exporter of wact-190 and other crystalizing molecules.

## PGP-14 is expressed in cells that surround the pharyngeal cuticle

We examined the spatial expression pattern of *pgp-14* in two ways. First, we mined the Cao *et al* L2 single cell sequencing dataset that provides detailed spatial information on gene expression [35]. Gene expression is reported based on the number of reads per gene in a given

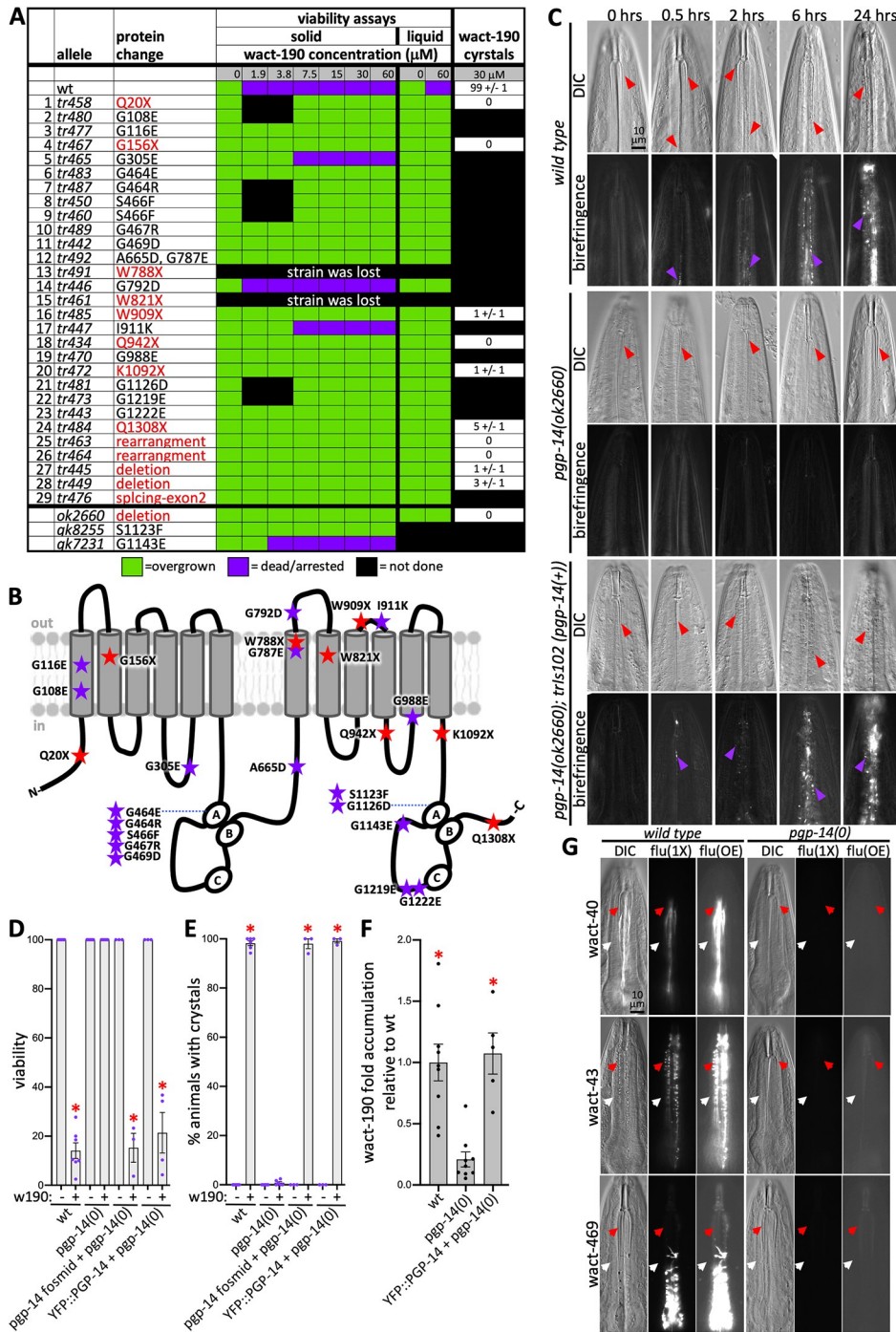

**Fig 3. Mutations in *pgp-14* Confer Resistance to wact-190 Crystals and Lethality. A.** A chart showing the 29 *pgp-14* alleles isolated in the forward genetic screen for resistance to wact-190 lethality. The behaviour of three independently isolated *pgp-14* alleles are indicated at the bottom of the chart. Select mutants were also assayed for their ability to form wact-190 crystals. Standard error of the mean is shown. **B.** A schematic of the locations of the indicated *pgp-14* alleles. **C.** A time course analysis of wact-190 crystal formation. The genotype for each series of animals is indicated on the left with DIC images on top with the corresponding birefringent images below the corresponding brightfield image. Time in hours (hrs) is indicated at the top. The sampling and photography of animals is terminal, hence a different representative animal is chosen for each time point. Red arrowheads indicate channels, purple arrowheads indicate birefringent crystals. The scale in the upper left panel is representative of all images in the series. **D-E.** Viability and crystal formation in the indicated genotypes in the background of solvent control (represented with a (-)) or 30 μM wact-190 (represented with a (+) w190). The *pgp-14*-containing fosmid used is WRM065dH09. YFP::PGP-14 is a

functional tagged PGP-14 coding sequence whose expression is driving by the *pgp-14* promoter (see methods for details). An asterisk indicates a significant difference ($p<0.001$) relative to the *pgp-14(0)* + wact-190 sample using a Student's T-test. **F.** Mass-spectrometry analyses of wact-190 accumulation in animals of the indicated genotypes. The number of independent repeats is shown as well as the mean and statistical differences using a Student's T-test. In D-F, the standard error of the mean is shown. **G.** The accumulation of three fluorescent (flu) wactives that form objects in association with the cuticle fail to obviously increase in abundance in the pharynx tissue in the *pgp-14(0)* mutant. The red arrowheads indicate the channels as a reference point; the white arrowheads indicate the pharynx tissue surrounding the channels. The third panel in each series is over exposed (OE) by 75% each relative to the unadjusted (1X) exposure time for each of the three wactive molecules. The scale in the left panel is representative of all images in the series.

cell type per 1 million reads overall. *pgp-14* is robustly expressed in the pharyngeal epithelial cells that include marginal, e epithelial, and arcade cells (a total of 16918 reads) [16,35,36] (Fig 4A and 4B). The *pgp-14* expression pattern is coincidental with established markers of the pharyngeal epithelium that include *hsp-43* (Fig 4C), *tat-3*, *ajm-1*, and *marg-1* (Fig 4F), but is not coincidental with markers of pharyngeal muscle (such as *mlc-1*, *mlc-2*, *myo-1*, and *myo-2* (Fig 4D and 4F) or the pharyngeal gland (such as *phat-1* and *phat-2*) (Fig 4F) [1,16,37]. *pgp-14* expression in all other cell types combined is low (502 reads total) [16,35]. Remarkably, *pgp-14* is the highest-expressed gene in pharyngeal epithelial cells, the second-highest expressed gene in the pharynx and is the 156[th] highest expressed gene when all cells of the animal are considered [35]. This *in silico* spatial expression pattern is consistent with a previous report of *pgp-14*'s expression pattern found through a systematic analysis of *C. elegans* PGPs [38].

To complement the *in silico* analyses of *pgp-14* expression, we built a functional reporter for *in vivo* expression analyses. The pPRZH1144 construct encodes YFP fused to the N- terminus of PGP-14, is driven by 1.6 kb of sequence upstream of the *pgp-14* start codon and is functional (see Fig 3D and 3E). We refer to the protein expressed from this construct as YFP::PGP-14 and find it to be expressed exclusively in the cells that surround the anterior pharyngeal cuticle, including the mc1 marginal cells and in the cells that surround the buccal cavity that likely include the arcade, e epithelial and pm1 and pm2 cells (Figs 4G, S4 and S5). The YFP::PGP-14 protein is enriched along the apical side of the marginal cells and buccal epithelial cells that surround the pharyngeal cuticle (red arrows and orange arrows, respectively, in Figs 4G and S5). We performed a time-course to determine whether PGP-14 localization changed in any way during the L4 to adult molt (see methods). We observed a striking increase in YFP::PGP-14 abundance in molting animals, much of which was also localized to the basolateral side of the marginal cells that surround the channel cuticles (compare the patterns indicated with the pink arrows in Fig 4G with that of Figs 4H and S5). A model of PGP-14 localization is presented in Fig 4I. The spatial localization of the tagged protein and its upregulation during molting suggests that PGP-14 may play a role in pharyngeal cuticle development.

## The spatiotemporal expression patterns of PGP-14 and SMS-5 overlap

PGP-14 was identified (but not described) in the same screen in which we identified the sphingomyelin synthase SMS-5 and related pathway components, suggesting that the two gene products may have related functions. We therefore investigated whether the spatial expression pattern of PGP-14 and SMS-5 overlap. Analysis of published single cell sequence data [35] indicates that while *sms-5* is expressed less abundantly than *pgp-14* (Fig 4E), it has the 90[th] most similar expression pattern to *pgp-14* in the pharynx relative to all other genes in the genome (Fig 4F). This is consistent with the expression pattern of the functional SMS-5::mCherry fusion protein that we previously described [20].

We next examined the overlap in the subcellular localization patterns of the functional SMS-5::mCherry and YFP::PGP-14 fusion proteins. We found some overlap between the two

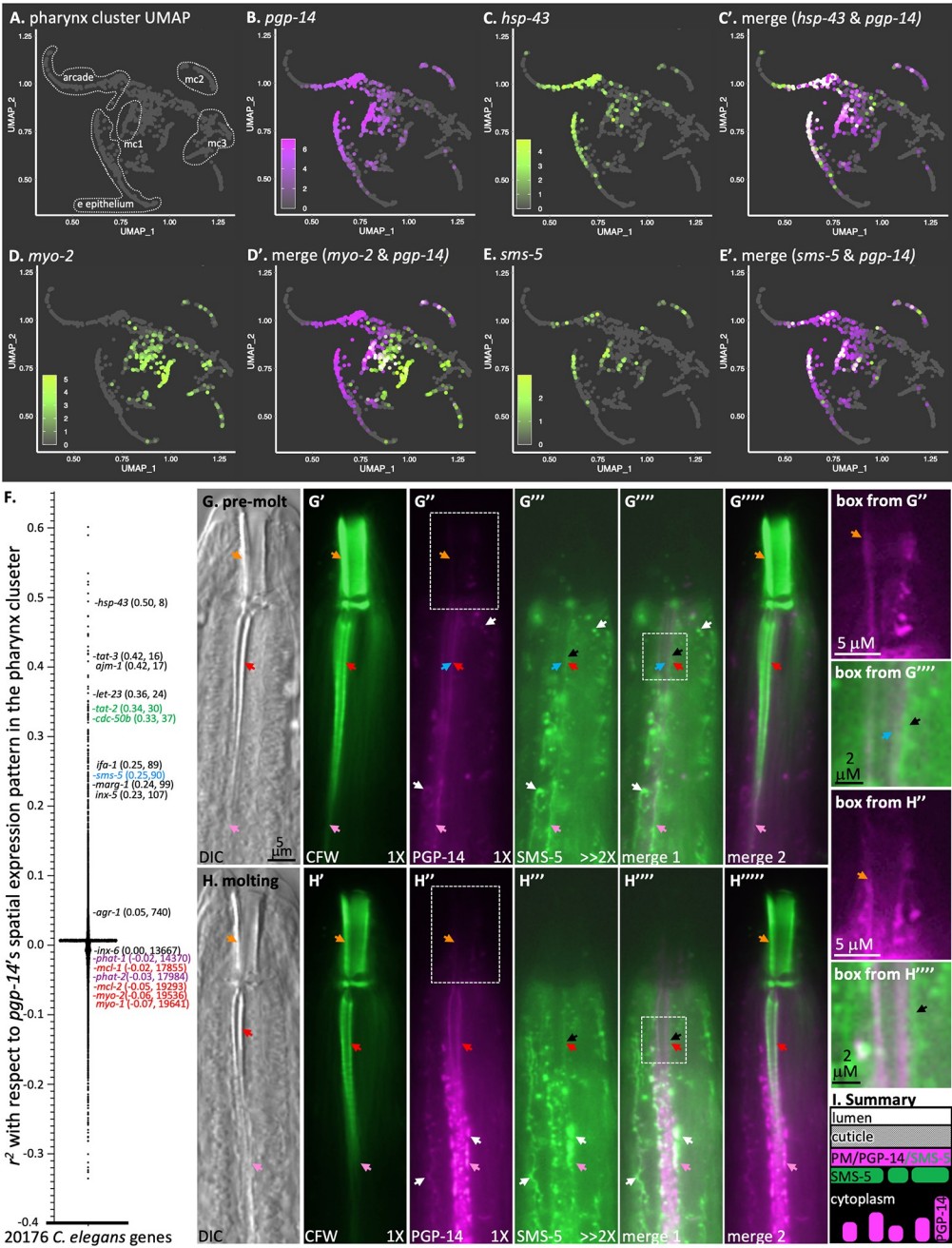

**Fig 4. PGP-14 is Expressed in the Anterior Pharynx. A.** A pharynx-restricted Uniform Manifold Approximation and Projection (UMAP) reference plot of pharynx cells mined from the Cao *et al* single cell *C. elegans* sequence analysis [35]. The identity of the epithelial clusters that surround the anterior pharynx cuticle are encircled in white [16]; mc1, 2, and 3 represent the marginal cells 1, 2, and 3 clusters respectively. **B.** The UMAP expression profile of *pgp-14*. **C&C'.** The overlay of *pgp-14* and the gold standard pharynx epithelial marker *hsp-43* [16]. **D&D'.** The overlay of *pgp-14* and the gold standard pharynx muscle marker *myo-2* [16]. **E&E'.** The UMAP expression profile of *sms-5* overlapped with that of *pgp-14*. In B, C, D, and E, the inset shows the expression level of the indicated gene as the log2 normalized transcript count per cell. **F.** A plot of the pharynx UMAP correlation values between *pgp-14* and all other genes in the *C. elegans* genome. Pharynx gold standards are indicated along with the respective Pearson correlation value with *pgp-14* and the overall rank of similarity to the expression pattern of *pgp-14*. **G.** The head of an RP3510 *trEx1010* [pPRHZ1138(*pgp-14p*::SMS-5::FLAG::mCherry); pPRHZ1144(*pgp-14p*::YFP::PGP-14)] animal before the L4 to adult molt (50 hours after synchronization at 20°C). **H.** The head of an RP3510 during the L4 to adult body apolysis (molting) stage where body cuticle shedding was evident (53 hours after synchronization). DIC, differential interference contrast; calcofluor white (CFW) and the functional tagged SMS-5 are shown in green; functional tagged

PGP-14 is shown in fuchsia. Merge 1 is the merge of the PGP-14 and SMS-5 micrographs, and merge 2 is the merge of the PGP-14 and CFW micrographs. The highlighted boxes on the right of the figure are magnified views of regions highlighted with a white box in the respective panels but with brightness, contrast and sharpness increased. The black arrows in the highlighted boxes from G""and H""highlight the juxtaposition of the tagged PGP-14 and SMS-5. As the channel travels from anterior to posterior, it veers closer to the central lumen. Hence, in a single focal plane, the apical side of the channel-surrounding marginal cells are in focus anteriorly (red arrows), while the basolateral side of the same marginal cells are in focus more posteriorly (pink arrows). Orange arrows indicate the buccal cavity cuticle, and the white arrows indicate the signal overlap between PGP-14 and SMS-5 that can be seen at the presumptive plasma membrane. The brightness and contrast of CFW and the tagged-PGP-14 signal before and during the molt are not manipulated (hence the 1X notation), but the lowly-expressed tagged-SMS-5 signal has been digitally increased (hence the >2X notation). The abundance of tagged PGP-14 increases obviously in molting animals and accumulates in packets basolaterally. An additional example, along with quantification, is shown in S5 Fig. **I.** A summary of the subcellular localization patterns with respect to the cuticle and the plasma membrane (PM).

proteins at the presumptive plasma membrane (light blue arrows in Figs 4G, 4H and S5). We also observed strong SMS-5 signal in vesicle-like structures, some of which abut tagged PGP-14 along the apical membranes that line the cuticle (black arrows in Fig 4G and 4H). While it is clear that PGP-14 and SMS-5 are expressed in many of the same cells and that there is some overlap in their subcellular localization, it is also clear that the overlap is incomplete.

We recently assembled a spatiotemporal map of pharynx gene expression [16]. We assembled this map by combining the RNA seq data from a detailed 16-hour time course during larval development [39] and single cell seq from 50,000 cells, also during larval development [35]. Remarkably, we find that both *pgp-14* and *sms-5* peak in expression within minutes of one another. The two genes peak just after what we have previously referred to as the 'peak molting hour' (Fig 5 and S1 Data). In general, there is a reasonable correlation between mRNA expression levels and protein abundance in *C. elegans* ($R_S$ of +0.59 [40]). Regardless, the peak in *pgp-14* mRNA expression is consistent with the increase in abundance of tagged-PGP-14 protein during the molt (Figs 4G, 4H and S5). Of note, we also find other genes that encode lipid-metabolism components peaking in expression around the same time as *pgp-14* and *sms-5*. These include *tat-2* and *cdc-50.b*, which encode the homologs of the ATP8B1 lipid flippase and its cofactor CDC50 [41] (Fig 5). Together, these observations are consistent with the idea that PGP-14 and SMS-5 coordinately play a role in the development of the pharyngeal cuticle.

## Nile red staining of the pharyngeal cuticle is dependent on PGP-14 and SMS-5

PGP-14 is homologous to ABCB4, which flops phospholipids from inner leaflet of the plasma membrane to the outer leaflet, or to an extracellular acceptor like bile acid [42,43]. We therefore asked whether PGP-14 might be playing a similar role in exporting phospholipids into the pharyngeal cuticle. We stained animals with Nile Red (NR), a lipophilic dye that fluoresces red in the presence of polar lipids [44,45]. We found that both *pgp-14* and *sms-5* null mutants have obviously less NR signal associated with the cuticle compared to the wild type (Fig 6A–6C). Coincidentally, there is obviously more NR signal in the tissue that surrounds the cuticle (Fig 6A–6C). By contrast, the patterns of calcofluor white (CFW) (a chitin stain) and Congo red (CR) (an amyloid stain) dyes that we have previously worked with [16] do not obviously change in the mutants (Fig 6D and 6F). We found that transgenically-expressed wild type *pgp-14* can rescue the NR deficit of *pgp-14(0)* mutants (Fig 6G and 6H). These observations suggest that both PGP-14 and SMS-5 are necessary for polar lipid accumulation within the pharyngeal cuticle.

## PGP-14 may flop phospholipids *in vivo*

Given the homology between PGP-14 and ABCB4, we asked whether PGP-14 might flop PCs *in vivo*. We previously demonstrated that *sms-5(0)* has a deficit of clustered sphingomyelin

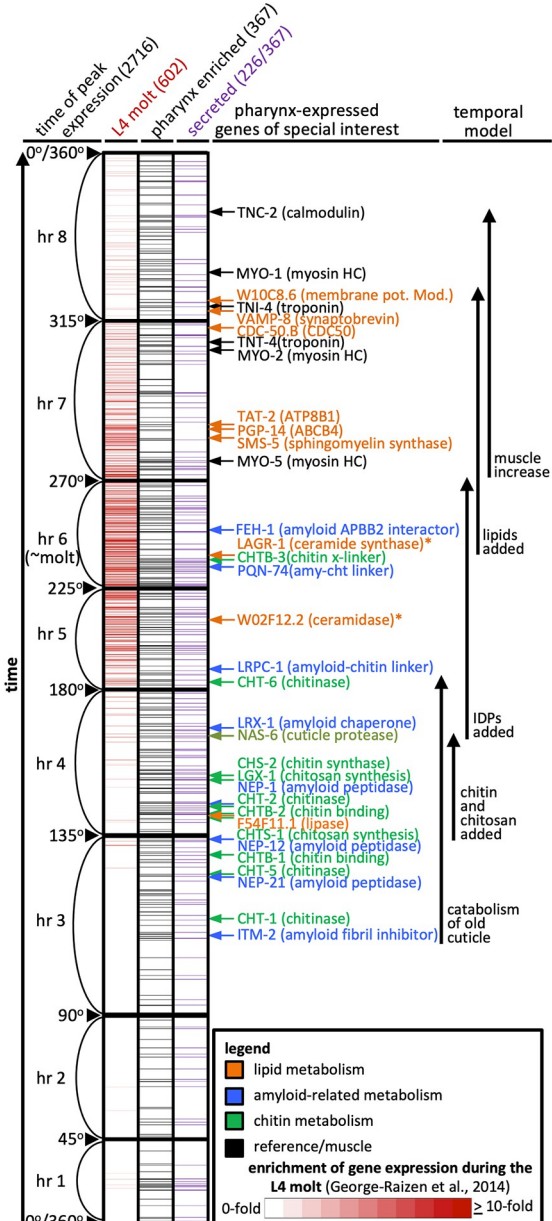

**Fig 5. A Temporal Expression Map of Genes whose Protein Products are Predicted to be Secreted and/or Expressed by the Pharynx.** The chart of the 2716 genes whose expression oscillates in expression level over the course of each larval stage is based on our meta-analysis [16] of previous work [39], but includes new highlights to show the peak expression of genes implicated in lipid-metabolism (in orange). The 2716 genes shown have expression levels that oscillate over each larval stage, peaking at the same relative time at each stage. Each row represents a single gene. The 2716 genes are arranged along the y-axis in order of the time at which each gene reaches its peak expression level with those earliest in the period at the bottom and those latest in the period at the top along the y-axis. The periodicity of oscillating gene expression is a continuum during larval development and time has previously been represented as degrees of a circle [39]. We have preserved that concept here and have indicated the degree along the y-axis, which is divided into bins of time relative to the molting period. The first data column (red) represents the expression level of the 602 genes on this chart that are upregulated during the L4 molt relative to surrounding timepoints (data from [15]; the scale is indicated in the legend). The second data column (black) represents the 367 genes from the set of 2716 that are enriched in expression in the pharynx (data from [35]). The third column (purple) shows the 226 genes (of the 367 pharynx-enriched set) that are predicted to be extracellular. Pharynx-expressed genes of special interest are indicated. The colour of the arrows and text corresponds to broad categories indicated in the legend. Our interpretation of how pharynx gene expression might impact pharynx cuticle development is shown on the right-hand side (IDPs, intrinsically disordered proteins). All details from this chart are available in the S1 Data.

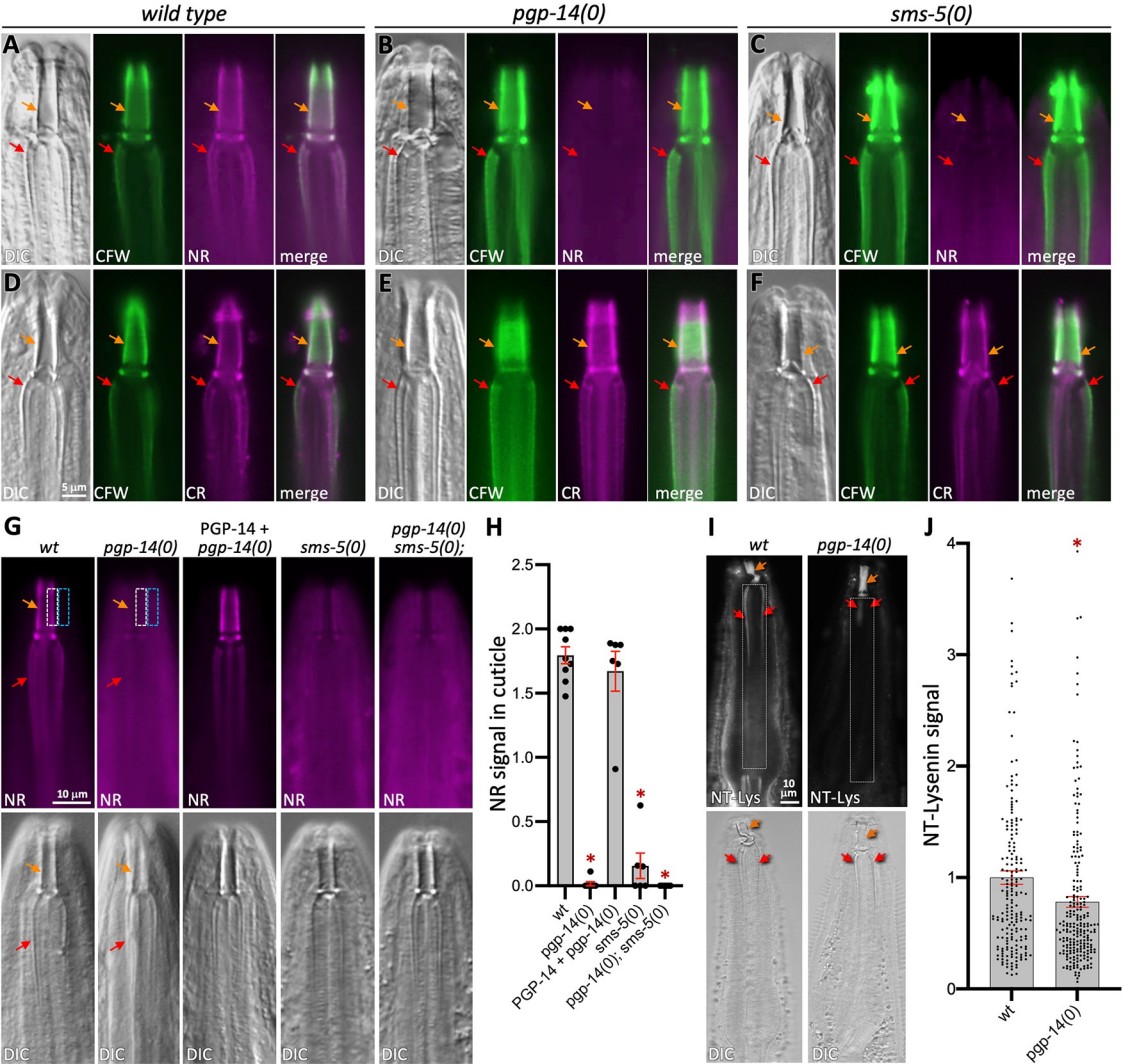

**Fig 6. *pgp-14* Mutants Have Reduced Nile Red and GFP-NT-Lysenin Signal within the Pharyngeal Cuticle. A-C.** Calcofluor White (CFW) and Nile Red (NR)-stained animals of the indicated genotype. **D-F.** CFW and Congo Red (CR) staining of animals of the indicated genotype (top). The scale bar in the upper left is representative of all images. At least 8 animals were examined in detail over the course of at least two independent trials and results were similar across replicates. DIC, differential interference contrast. **G-H.** Additional analyses of the deficit of NR signal in the cuticle of animals with the indicated genotypes (top). The NR signal in cuticle shown in H is calculated by subtracting the background signal (blue boxes shown in the two left-most panels as examples) from the signal associated with the cuticle (white boxes shown in the two left-most panels as examples). **I.** An analysis of *wild type* and *pgp-14(0)* null animals stained with the GFP-NT-Lysenin protein probe (NT-Lys) that recognizes clustered sphingomyelin. The fluorescent image is on top and the respective DIC image is below. The scale indicated in the upper left is the same for all panels. **J.** Quantification of GFP::NT-Lysenin signal whereby each dot represents an independent measurement of the average signal intensity of the area outlined in the white boxes relative to an equivalently sized area next to the animal on the slide. Two outliers for wt and one for *pgp-14(0)* are not shown to provide added clarity to the graph. The *pgp-14(0)* mutant exhibits significantly less staining ($p = 0.004$) than wild type by a Student's T-test. For both graphs, the mean and standard error of the mean is shown. In all panels, the orange arrows indicate the buccal cavity cuticle and the red arrows indicate the channel cuticle.

(SM) on the surface of the pharynx epithelium [20] using a protein probe called GFP::NT-lysenin [46,47] on animals whose cells were not permeabilized. The active site of the SM synthases resides on the outer leaflet of membranes where it uses PC as a substrate to make SM [48]. If PGP-14 flop PCs to the outer leaflet, then *pgp-14(0)* might also have a deficit in SM abundance as detected by a decreased GFP::NT-lysenin signal. Indeed, we observe a significant decrease

in GFP::NT-lysenin signal in the *pgp-14(0)* mutant background (*p* = 0.004; Fig 6I and 6J). The fold decrease in clustered SM in *pgp-14(0)* is not as dramatic as that observed in the *sms-5* null background [20]. These observations suggest that SMS-5 uses other substrates beyond what PGP-14 may provide, which is not unexpected. Regardless, this data is consistent with the hypothesis that PGP-14 may flop PC *in vivo*.

## PGP-14 and SMS-5 contribute to a xenobiotic barrier

Cuticle lipids are known to act as xenobiotic barriers in insects [49,50]. Hence, lipid content within the pharyngeal cuticle might not only act as a sink for select small molecules like wact-190, but might also be a barrier to xenobiotic entry into the cells of the animal. We tested this hypothesis in two ways.

First, we asked whether *pgp-14(0)* mutants are more permeable to the Hoechst 33342 dye. This fluorescent dye binds to DNA and can permeate the plasma and nuclear membranes [51]. Previous work demonstrated that Hoechst 33342 can penetrate the *C. elegans* exocuticle and eggshell only when their respective lipid layers are depleted [6,8,52]. We find little-to-no Hoechst 33342 signal in the heads of wild type animals, but obvious nuclear signal in the heads of *pgp-14(0)* mutants, which is rescued by the *pgp-14(+)* transgene (Fig 7A–7D).

Second, we asked whether *pgp-14(0)* mutants might be hypersensitive to bioactive small molecules. Similar to our previous work with the *sms-5(0)* null mutant [20], we examined *pgp-14(0)* sensitivity to 508 molecules of our worm-active library [53]. We found that 140 molecules (28%) kill *pgp-14(0)* mutants at some concentration that does not kill the wild type (Fig 7E and S1 Data). The *sms-5(0)* mutant is hypersensitive to 139 of these 140 molecules (99%) plus an additional 48 molecules that kill wild type and *pgp-14(0)* mutants equally (Fig 7E and S1 Data). Together, we refer to these 188 molecules as 'hypersensitive' molecules.

Our chemical-genetic interaction matrix reveals that 28 molecules kill wild type animals at some concentration that fail to kill the *pgp-14(0)* mutant strain (Fig 7E). *sms-5(0)* mutants are also resistant to 27 of these 28 molecules (96%) in at least one of the concentrations tested (Fig 7E). For simplicity, we refer to these compounds as 'resistant' molecules. These 28 molecules include the aforementioned crystalizing molecules wact-190 and wact-469. Of the 28 'resistant' molecules, 26 have been examined for their ability to form objects in the heads of worms at a concentration of 30 μM ([20] and this work (S1 Data), and 23 (88%) form birefringent crystals, with another two forming non-spherical objects that may be crystal-precursors. This suggests that the lipid barrier within the cuticle may serve as a sink for other crystalizing molecules beyond wact-190 and wact-469.

The pattern of interactions between the *pgp-14(0)* mutant with the worm-active library is more similar to that of *sms-5(0)* than the wild type (*p*<0.001; see $r^2$ scores at the top of Fig 7E). Given that the phenotypes and expression patterns of *pgp-14* and *sms-5* mutants are so similar, we asked whether they are likely to function in the same pathway. Should PGP-14 and SMS-5 activity be restricted to the same biochemical pathway, then the *pgp-14(0); sms-5 (0)* double mutant should not exhibit a phenotype that is more severe than either single mutant. We performed this genetic test by measuring the sensitivity of wild type animals, the double mutant and both single mutants in dose-response assays with four hypersensitive molecules. We found that the EC50 of the double mutant decreased significantly (*p*<0.05) relative to the two single mutants for two of the four molecules tested (Fig 7F). This mild phenotypic enhancement suggests that there may be other lipid-synthesis enzymes and PGPs that work in parallel to SMS-5 and PGP-14 that make minor contributions to the pharyngeal cuticle lipid barrier.

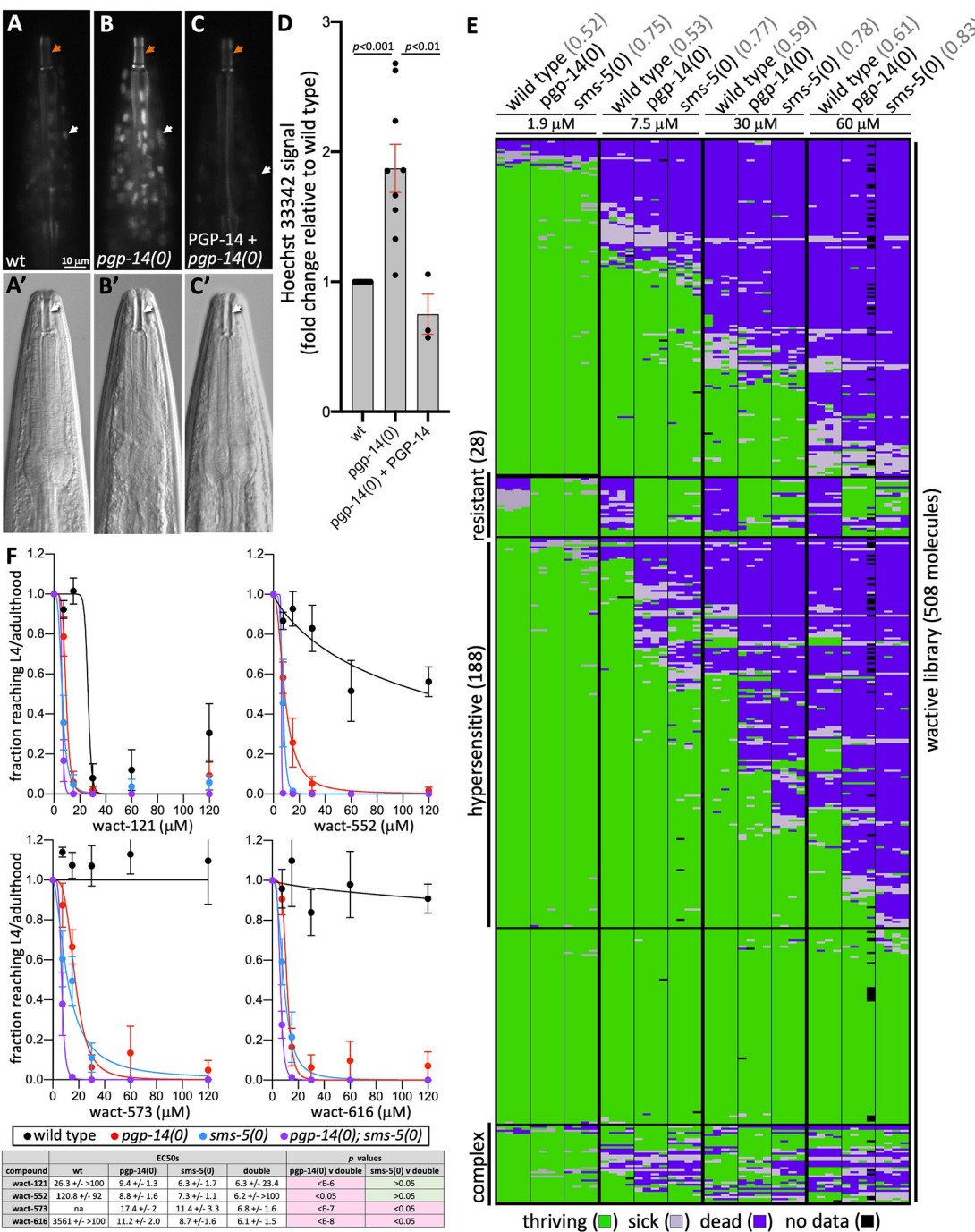

**Fig 7. *pgp-14* Mutants Have Altered Xenobiotic Barrier Properties. A-C**. Heads of animals stained with 5 ug/ml Hoechst 33342 (see methods). The top shows the Hoechst signal and the bottom shows the corresponding DIC images. The rescuing PGP-14 array is *trIs102*[pPRHZ1143(*pgp-14p::pgp-14*(genomic/cDNA)::unc-54(3'UTR)); pPRHZ1131(*pgp-14p*::YFP)::YFP)]. Orange arrows indicate the buccal cavity, white arrows indicate the Hoechst-stained nuclei. **D.** Quantification of Hoechst signal over multiple trials, for which each trial is represented by a dot and each trial includes at least 10 different animals. **E.** Interactions of strains of the indicated genotype (x-axis) with molecules from our wactive small molecule collection (y-axis). *pgp-14(ok2660)* and *sms-5(ok2498)* null mutants are used. Concentrations of the wactives are indicated in the column headers. Each strain was tested against each molecule four times. The correlation of interactions between the *pgp-14(0)* mutant and either wild type or *sms-5(0)* is indicated in gray in line with the respective notation. **F** Dose response analyses of four 'hypersensitive' molecules (i.e., molecules whose potency in the mutants is higher relative to wild type) in the indicated genetic background. The graphs show the means and standard error over four independent trials (N = 4) with four technical replicates each (n = 4). The table shows the EC50s, 95% confidence intervals for the dose response for the four molecules shown on the graph. Statistical tests (using an extra sum-of-

squares F test) reveals that the double *pgp-14(0); sms-5(0)* mutant is significantly more sensitive to wact-573 and wact-616 compared to the most sensitive single mutant (*sms-5(0)*) ($p<0.05$). All calculations were performed using GraphPad Prism (version 9.3.1).

## Discussion

Multiple lines of evidence support a model whereby PGP-14 creates a barrier to xenobiotic insult by facilitating polar lipid deposition into the developing pharyngeal cuticle of *C. elegans* (Fig 8). First, PGP-14 is spatially localized to the apical membrane that abuts the anterior pharyngeal cuticle. Second, *pgp-14* expression peaks during the time of new cuticle construction and coincidentally with other lipid interactors like SMS-5 and TAT-2. Third, staining of the pharyngeal cuticle by the polar lipid dye Nile Red is reduced in *pgp-14* nulls. Fourth, the loss of PGP-14 reduces the abundance of clustered sphingomyelin within the apical membrane of the anterior pharynx. Less sphingomyelin is indicative of a phosphatidylcholine deficiency on the outer leaflet of the plasma membrane [46]. Fifth, the penetration of the Hoechst 33342 dye, which is known to be retarded by the lipid within cuticles [6,8,52], is markedly increased in the absence of PGP-14. Finally, animals lacking PGP-14 are hypersensitive to effects of many compounds. Given the many natural products that nematodes encounter in the wild, a lipid barrier within the pharynx cuticle is likely an important feature that promotes survival.

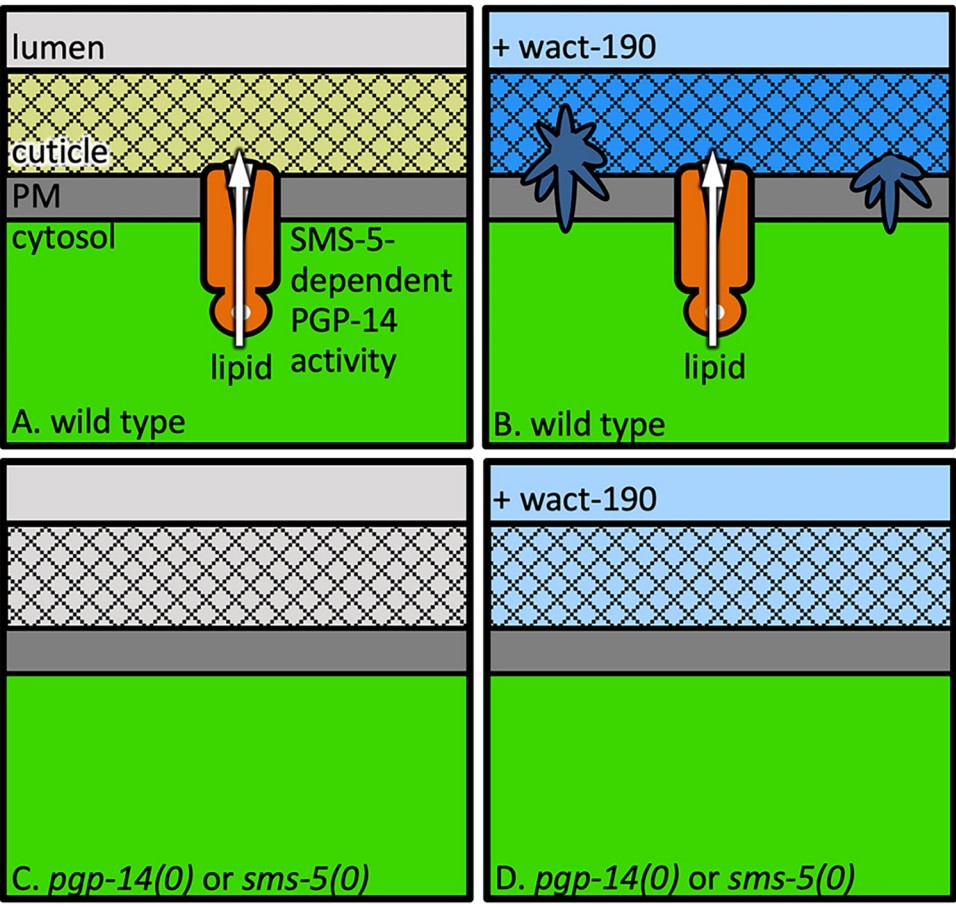

**Fig 8. A Model of PGP-14's Role in Pharyngeal Cuticle Development and Xenobiotic Crystal Formation.** Light blue indicates soluble xenobiotic (the wact-190 small molecule) passing through the alimentary tract; medium blue indicates soluble wact-190 accumulation in the pharynx cuticle; dark blue indicates crystalized wact-190.

The lipid content within a cuticle can fulfill a barrier role likely because its hydrophobicity prevents the free passage of hydrophilic compounds. Paradoxically, the PGP-14-mediated polar lipid deposition in the anterior pharynx inadvertently acts as a sink for the accumulation of hydrophobic small molecules. We speculate that the amyloid-like material within the pharynx cuticle [16] may seed the formation of crystals from the concentrated small molecules, depending on their physicochemical properties. We suspect that crystal formation is unlikely to occur in the wild given that we observe it in the laboratory at micromolar concentrations. Natural products are unlikely to be common at such high concentrations in the wild. Lipid barriers are therefore unlikely to drive small molecule crystallization in the worm's natural habitat.

Like all *C. elegans* PGPs, PGP-14 is a member of the ABCB class of whole transporters [23] and is most homologous to human MDR1 (ABCB1) and MDR3 (ABCB4) (see S1 and S2 Figs). MDR1 is the primary drug efflux pump within human cells [54] and can be mutated to confer multi-drug resistance under negative selection pressure [55,56]. An obvious alternative model to explain PGP-14's role in crystal formation is that it pumps crystallizing molecules from the pharynx epithelium, which then precipitate out of solution once concentrated outside of the epithelium. Also in this model, the 188 'hypersensitive' molecules (Fig 7E) are those that are normally pumped out of the pharynx epithelium by PGP-14, but hyperaccumulate in the *pgp-14* mutant.

We do not favour this alternative model for multiple reasons. First, fluorescent molecules that fail to crystalize in the *pgp-14(0)* mutant do not obviously accumulate in the cells that normally express PGP-14 (or any other cells) (see Fig 3G). The same is true for the lack of accumulation of wact-190 in the *pgp-14(0)* mutant relative to the wild type as detected by mass-spectrometry (see Fig 3F). The molecules that crystalize are hydrophobic, and worms cultured in water-based media should act as a sink for their accumulation. These molecules should be readily observable in the *pgp-14(0)* mutant if PGP-14's role is to export these molecules, but they are not. Second, in *mlt-9(RNAi)* animals, the crystals are tightly associated with the detached pharynx cuticle (see Fig 2T). If the crystallizing molecules were exported from epithelium by PGP-14, the resulting crystals might be randomly distributed within the space created by the detached cuticles of *mlt-9(RNAi)* animals, but that is not what is observed. Third, under this alternative model, the crystallizing molecules might be expected to be more hydrophobic than the 188 hypersensitive molecules. Otherwise, the hypersensitive molecules might also precipitate out of solution once exported from the epithelium. We see no statistical difference in the hydrophobicity between the two groups of molecules ($p > 0.23$; S1 Data). Fourth, fewer than 10% of crystal-forming compounds are predicted to be MDR1 substrates (S1 Data). Finally, unlike canonical multi-drug export pumps, there is little evidence to suggest that *pgp-14* expression is xenobiotic-responsive. Despite *pgp-14* being expressed at higher basal levels (along with many other genes) in a strain evolved to resist ivermectin [57], large-scale systematic expression analyses show that *pgp-14* expression is not co-regulated with xenobiotic detoxifying enzymes like *pgp-1, -2, -3, -5, -6, -8, -9* and *pgp-13* [58]. Instead, *pgp-14* is co-regulated with genes involved with pharynx function such as *myo-1* and *myo-2* [58]. Consistent with these findings, a second study demonstrated that the upregulation of *pgp-14* expression is independent of a major xenobiotic-responsive nuclear receptor, NHR-8, unlike several other *pgp* genes that are NHR-8-dependent [59]. The upregulation of *pgp-14* in the aforementioned evolved strain may contribute to a fortified lipid barrier or may simply be inconsequential. For some of these points, we can conceive of different explanations that would account for the observations under a drug-pump model, but these explanations are non-parsimonious. For these reasons, we favour the model that PGP-14 contributes to a polar lipid barrier, which parsimoniously accounts for all of the observations.

As introduced above, apical ABCB4 plays a key role in bile production by flopping phosphatidylcholine (PC) phospholipid from the inner leaflet of the plasma membrane to the outer leaflet of hepatocytes [42,43,60]. There, bile salts within the canalicular lumen act as acceptors of the PCs to form micelles that are used to digest fats [61–63]. ABCB4's interaction with PC has recently been characterized by cryo-EM, which revealed four ABCB4 residues required for PC interaction [64]. No *C. elegans* PGP has more than one of these four residues conserved (see S1B–S1D Fig), suggesting that nematode PGP interaction with phospholipids may have diverged to maintain PC homeostasis within the plasma membrane. While our NT-lysenin results suggest that PGP-14 may flop PCs to the outer leaflet along the pharynx plasma membrane (see Fig 6J), it is entirely possible that PGP-14 exports other phospholipids, which may or may not include PCs, to the developing pharynx cuticle. Developing methods to purify intact pharynx cuticle for lipidomic analyses in the future will help elucidate the nature of PGP-14's *in vivo* substrate.

PGP-14 and SMS-5 clearly play an overlapping role in the same biological process. *pgp-14* and *sms-5* were identified in the same genetic screen for mutants that can resist crystal-associated death. Both mRNAs peak in expression at the same time during cuticle construction, and both are expressed in the epithelium that produce the pharynx cuticle. Both *pgp-14(0)* and *sms-5(0)* mutants exhibit the same Nile Red phenotype within the pharynx cuticle, and both mutants show a similar pattern of chemical-genetic interactions with the wactive small molecule collection.

Despite these similarities, the evidence suggest that PGP-14 and SMS-5 do not function as an obligate pair. First, while their spatial expression patterns and subcellular localization overlap, the overlap is not complete. SMS-5 is expressed in pharynx cells and the spermatheca where PGP-14 is not expressed. While their pattern of chemical-genetic interactions are positively correlated (*pcc* >0.75; see Fig 7E), the pattern of interactions is not identical. While both mutants show a significant loss of clustered sphingomyelin in the pharynx, *pgp-14(0)*'s sphingomyelin-related phenotype is not as extensive as *sms-5(0)*'s. Finally, and perhaps most telling, double mutant analyses suggest that the activities of the two gene products are not restricted to one another's pathway. This suggests that there may be other PGPs that work with SMS-5, and other sphingomyelin synthases that function with PGP-14. Given that mutations of other PGPs and SMSs do not have *pgp-14(0)*-like phenotypes (see S3 Fig and [20]), these other components likely make only minor contributions to PGP-14 and SMS-5 activity.

Several models can explain how SMS-5 may function with PGP-14 to establish a lipid barrier within the pharyngeal cuticle. One model is that SMS-5-generated sphingomyelin is exported by PGP-14 into the cuticle. However, we see this as unlikely for two reasons. First, the active site of characterized sphingomyelin synthases resides on the outer leaflet of membranes [48], which is incongruent with the known roles of P-glycoprotein proteins, which pump substrates from inside the cell to the outside of the cell [65]. Second, in our analysis of Nile Red staining, the loss-of-function of both PGP-14 and SMS-5 results in more Nile Red signal in the interior of the animal compared to the wild type (see Fig 6). If SMS-5-generated sphingomyelin was the PGP-14 substrate, we might expect an absence of Nile Red signal within the pharynx epithelium in the *sms-5(0)* mutant, which is not what we observe. Instead, we favor a model whereby PGP-14 function is largely dependent on SMS-5. That is, PGP-14 must reside in a sphingomyelin-rich environment to function efficiently, which is consistent with the known sphingomyelin-dependance of ABCB4 [66]. In this model, PGP-14 exports a non-sphingomyelin polar lipid into the developing pharyngeal cuticle (Fig 8).

PGP-14 is not the first ABC family protein implicated in cuticle lipid barrier formation. Disruption of ABCH-9C in *Locust migratoria* reduces cuticle lipid content and makes the insects hypersensitive to desiccation and small molecule penetration [49]. In the fruit fly *D.*

*melanogaster*, the ABCG family member *Snu* helps transport a lipid-protein mixture called cuticulin into the developing cuticle. Like ABCH-9C, loss of *Snu* makes flies hypersensitive to water loss and xenobiotic attack [50]. ABCG family members have also been implicated in transporting cuticular lipids in several species of plants [67–69]. Hence, ABC family members play important roles in establishing cuticle lipid barriers in a wide variety of life forms.

The similarities between *C. elegans* PGP-14 and human ABCB4 biology is remarkable in many respects. First, the two gene products likely have similar molecular functions in that PGP-14 plays a role in polar lipid deposition into the pharynx cuticle of *C. elegans* and ABCB4 exports PCs into the bile canaliculus of mammals. Second, the two gene products function in analogous anatomical tissues in that both the pharynx channels and the liver canaliculus are side branches of the main digestive tract. Third, TAT-2 and CDC-50.b, which are the *C. elegans* homologs of human ATP8B1 and CDC50 that function in opposition to ABCB4 to regulate lipid homeostasis in the hepatocytic membrane [70], are highly enriched for expression within the pharynx and peak in expression at roughly the same time as PGP-14 during the molting cycle (see Fig 5). Finally, *pgp-14(0)* and *sms-5(0)* mutants have highly similar phenotypes while ABCB4 activity is dependent on sphingomyelin abundance within the plasma membrane [66]. Together, these similarities raise the possibility that PGP-14 function represents an ancient role for the ABCB4 family members within the digestive tract of the common ancestor between worms and humans.

## Methods

### Animal ethics statement

We (the authors) affirm that we have complied with all relevant ethical regulations for animal testing and research. Given that our experiments focused exclusively on the invertebrate nematode worm *C. elegans*, no ethical approval was required for any of the presented work.

### *C. elegans* culture and synchronization

*C. elegans* strains were cultured and synchronized as previously described [16,20]. Briefly, gravid adults were washed off plates with M9 buffer, centrifuged at 800 *g* to remove supernatant, followed by additional washes in M9 buffer until bacteria were removed. Worms were then collected as a 1.5 mL-pellet in a 15 mL conical tube. Next, 1 mL of 10% hypochlorite solution (Sigma) was added to the tube followed by 2.5 mL of 1 M sodium hydroxide solution and 1 mL double-distilled water, and the mixture was incubated on a nutator for 5 minutes. With 1.5 minutes remaining for incubation, the tube was vortexed for 10 seconds with two 5 second pulses after which the harvested eggs were washed four times with 12 mL M9 buffer, with vortexing after every addition of M9 buffer. After the final wash, the tube was incubated overnight on a nutator at 20˚C to allow eggs to hatch, soon after which the L1 hatchlings arrest their development until they are fed. and checked the next day for synchronized L1s. To obtain other synchronized stages, the synchronized L1s are plated on solid agar substrate with *E. coli* food and allowed to progress to the desired stage before processing. The forward genetic screen from which *pgp-14* alleles were isolated is described previously [20].

### *C. elegans* drug viability assays

Six-day viability assays were conducted as previously described [20]. Briefly, a saturated culture of HB101 *E. coli* was concentrated 2-fold in liquid Nematode Growth Medium (NGM) and 40 μL of bacterial suspension was dispensed into each well of a flat-bottom 96-well plate. Compounds and DMSO controls were pinned into the 96-well plates using a 96 pin replicator

with a 300 nL slot volume (V&P Scientific). The final concentration of DMSO in each well was 0.6% v/v. Approximately 20 synchronized L1 larval stage worms (see above) were added to each well of the 96-well plates in 10 μL of M9 buffer [71]. Assay plates were sealed with Parafilm and incubated for 6 days at 20°C with shaking at 200 rpm in a New Brunswick Scientific I26 incubator shaker. After 6 days, the plates were observed using a dissection microscope and the wells were categorized according to the number of viable adult and larval stage worms in each well. Wells with more than 50 animals after 6 days were categorized as over-grown. The number of worms in wells with approximately 50 or fewer worms were counted. All six-day viability assays were completed in quadruplicate.

The viability assay with the RP3192 strain carrying *pgp-14(ok2660)*; Ex[pPRGS382 (myo-2p::mCherry), pTG96(sur5p::GFP); WRM065dh09 (fosmid containing wild type *pgp-14*)] had to be carried out differently because of the mosaic nature of the inheritance of the Ex (extrachromosomal) transgenic array. On day 0, ten gravid adult worms carrying the Ex array (recognized by the expression of fluorescent reporters from the array) were plated on 6-cm plates containing either 30 μM wact-190 or DMSO solvent-only control and seeded with OP50 *E. coli* food. The adults were allowed to lay eggs for 6 hours and then picked off. On day 4, the number of F1 Ex(+) and Ex(-) animals that grew to L4 or adulthood were counted and expressed as a fraction (percentage) of total number of Ex+ and Ex- worms counted. The experiment was performed in three biological trials with 2–3 technical repeats with each technical repeat having more than 17 worms counted. The graph normalizes the data relative to the solvent control samples.

## Crystal analyses

Performed as previously described [20]. Briefly, synchronized L1s were added to wells of a 96 well-plate (50 L1s / well) containing liquid NGM, *E. coli* (HB101) as food source (OD600 = 1.2) and 30 μM of wact-190 in quadruplicate wells. 48 hours later, worms were transferred to Eppendorf tubes, and washed once with fresh M9 and spun down at 1800 x g for 1 minute to form a tight worm pellet of ~10 μL. Then, 5 μL of 50 mM levamisole (to a final concentration of ~ 16.7 mM) was added to paralyze worms. Worms were then mounted on a 2% agarose pad on a glass slide. Live worms were then observed for presence of birefringent crystals in the pharynx using 40 X objective of a Leica DMRA microscope. A minimum of three biological replicates were processed unless otherwise indicated, and at least 20 worms per replicate were counted for the presence or absence of spheres and/or crystals.

The crystal counts with the RP3192 strain carrying *pgp-14(ok2660)*; Ex[pPRGS382 (myo-2p::mCherry), pTG96(sur5p::GFP); WRM065dh09 (fosmid containing wild type *pgp-14*)] had to be carried out differently because of the mosaic nature of the inheritance of the Ex (extrachromosomal) transgenic array. The experiment was set up similar to that of the RP3192 viability test described above except that on day 2, F1 animals were mounted on a microscope slide and observed for wact-190 crystal birefringence in the pharynx of Ex+ and Ex- animals. The number of F1 Ex+ and Ex- animals that had wact-190 crystals in the pharynx was expressed as a fraction (percentage) of total number of Ex+ and Ex- worms that were looked at. The experiment was performed in three biological trials with 2–3 technical repeats with each technical repeat having more than 40 worms counted.

## Molecular biology and transgenics

The pPRHZ1144 (*pgp-14p*::YFP::PGP-14(genomic)::*pgp-14*(3'UTR)) construct was built using a PCR stitching strategy. First, the pPRHZ1160 construct was built, which is a 7.7kb SacI-NaeI cut fragment of the *pgp-14*-containing fosmid wrm06dh09 cloned into the pKS plasmid vector.

pPRHZ1160 harbours 1.8 kb of *pgp-14* promoter followed by the 5.1 kb of *pgp-14* exons, introns and 3'UTR from wrm06dh09. Next, the following three PCR products were generated, then spliced together in a final PCR reaction: i) From pPRZH1160, a PCR product of 1.5 kb upstream of the *pgp-14* starting codon was generated; ii) from Andy Fire's L4640 construct, a PCR product of 889 bp encoding YFP, was generated; iii) from pPRZH1160, a PCR product of 329 bp of the 5' *pgp-14* genomic sequence, starting with the start codon was generated. All three of these products were stitched together using PCR, then reinserted into pPRZH1160 to create pPRZH1144, which was sequence verified and shown to be functional by rescuing mutant *pgp-14*'s resistance to wact-190.

RP3510 harbours a high-transmitting extra-chromosomal array trEx1010 that was made by injecting wild type N2 worms with p1138(pgp-14p::SMS-5B(genomic)::FLAG:: mCherry) (5ng/ul), p1144(pgp-14p::YFP-PGP-14(genomic)::pgp-14(3'UTR))(5ng/ul), and pKS(90ng/ul).

## Analysis of PGP-14 and SMS-5 subcellular localization during the L4 to adult molt

The time course analysis of tagged-PGP-14 and SMS-5 expression within the RP3510 strain over the course of the L4 to adult molt began with L1s synchronized from an egg prep done the previous evening. L1s were plated on solid plates seeded with OP50 *E. coli* and incubated at 20°C. We began our inspection 44 hours later by incubating worms in Calcofluor white (CFW) for 1 hour, followed by rinsing and mounting on standard agar pads and anesthetized with 100 mM of sodium azide. Worms were inspected at 630X magnification on a Leica DRMA microscope approximately every two hours with a preceding 1 hour incubation with CFW until the majority of the population reached adulthood after the 52 hour time point as judged by fertilized embryos in their uteri. We note that RP3510 grows slower than wild type animals and animals become asynchronized over their 52 hour incubation. Individual animals were judged to be molting by the apolysis of their body cuticle (that is, the shedding of their old cuticle). Animals were judged to be adults by the presence of embryos in their uteri. Animals were judged to be pre-molt L4s by the absence of apolysis or embryos in their uteri.

The relative expression level of YFP::PGP-14 was quantified by first synchronizing an L1 population (as previously described [20]) of the RP3510 strain carrying the YFP::PGP-14-expressing extrachromosomal (Ex) array. Synchronized animals were incubated on MYOB plates with *E. coli* food for ~60 hours at 20°C. When the worms reached the L4 stage, they were imaged every ~2 hours (while being incubated at 20°C) by washing the animals in M9 and incubating them for 1 hour in 52 μM CFW to mark the cuticle, to catch intermolt L4 animals and then those L4s that had visible signs of molting (cuticle separation). The YFP fluorescence signal was quantified in the Ex positive animals along i) the anterior pharynx channels, and ii) the surrounding signal on the slide. For each animal, the background signal was subtracted from the channel signal for each worm. The background-corrected YFP::PGP-14 signal intensity was quantified using ImageJ default settings. For each animal, we expressed the YFP:: PGP-14 signal in molting and intermolt animals as the fold increase relative to the average signal from the intermolt animals.

## Generation of UMAP plots

UMAP plots were generated using the single-cell RNA sequencing (scRNA-seq) data set for L2 stage *C. elegans* that was previously published [35] and reanalyzed and re-projected by Packer *et al.* [72] with improved cell type annotations. We have previously further refined these cell type annotations [16]. We downloaded this reprocessed L2 stage scRNA-seq data from https:// github.com/qinzhu/Celegans.L2.Cello for our analyses. We analyzed this data using R and

used the ggplot2 package to visualize the Log2 normalized expression data for single genes of interest in each cell on the UMAP plot, as well as the co-expression of two genes simultaneously [73,74]. The pharynx-specific UMAP plots display only those cell clusters that contain pharyngeal and arcade cells as annotated based on marker gene expression classified previously [72]. The pharyngeal gland cell clusters (containing cell types g1A, g1P and g2) were omitted from these plots for clarity.

## Pulse chase experiment

We previously described our pulse-chase methodology [16]. Briefly, synchronized wildtype L1 worms are plated on 10cm plates at 7,000 L1s/plate on OP50. "L4" plates are incubated at 16˚C while "adult" plates are incubated at 20˚C (Day 0). On Day 3, the "L4" plate is taken out and stored at 20C overnight. On Day 4, "L4" and "adult" plates are washed with M9 to remove bacteria. Note that for the wact-469 pulse-chase experiment, wact-469 was coincubated with Eosin Y and the resulting Eosin Y images (but not wact-469) was published in [16]. For the duo Eosin Y + wact-469 incubation, 50uL of worm pellet was added to 430uL of NGM+ 15uL of 5mg/mL Eosin Y + 60uM wact-469 in 1% DMSO in 1.5mL Eppendorf tube (final volume 500uL). The tubes were incubated in a rotating nutator for 3hrs in the dark at room temperature. After incubation, the tubes were spun at 5000rpm for 1min to remove excess dye/drug after which the concentrated pellet was carefully transferred to 15mL falcon tube and washed with 8mL of M9 buffer. The tubes were inverted gently and spun at 2000rpm for 1min. The supernatant was removed and the concentrated washed worm pellet was spotted onto the clear (agar) surface of 6cm plates seeded with OP50. Worms were allowed to separate from residual dye/dead worms and debris onto the OP50 lawn, for 30min. 20–30 worms were then picked onto a second plate (35mm petri plate) lightly seeded with OP50. Immediately worms with staining and/or drug were visualized under epifluorescent microscope using the YFP and CFP filter sets and the numbers recorded. Note that the number of worms should be limited to 1000 because adding more worms reduces the number of worms that get stained. Also, the tip of siliconized tips were cut with flame-sterilized scissors to avoid injuring the L4/adult worms.

## Analyses of attached shed cuticle

We previously described our *mlt-9(RNAi)*-related methodology [16], which is modified from the work of Frand *et al* [27]. Briefly, a bacterial culture expressing dsRNA of *mlt-9* (referred to here as *mlt-9(RNAi)*) [75] was started from a single colony in 30 mL LB broth containing 100 μg/mL ampicillin for 18 hrs at 37˚C at 200 rpm. The cells were pelleted by centrifuging at 3200 rpm for 15 min, after which the cells were concentrated 10-fold. 1mL of the pelleted cells were added to 10 cm NGM agar plates containing 8 mM IPTG and 40 μg/mL carbenicillin and left to dry overnight at room temperature in the dark. The next day (day 0), 6,500 synchronized L1s were plated onto each RNAi plate, after which the plates were stored at 16˚C in the dark. Ninety hours later, the worms were inspected for *mlt-9* RNAi phenotypes. Approximately 50% of *mlt-9(RNAi)*-treated worms exhibit the expected cuticle defects. Performing mock RNAi with the empty L4440 plasmid failed to yield worms with obvious molting defects.

We incubated *mlt-9(RNAi)*-treated worms with wact-469 as follows: Worms were washed with M9 to remove bacteria after 90 hours of incubation with *mlt-9(RNAi)* at 16˚C. The worms were pelleted and 50 μL of the worm pellet (~20 worms/μL) was added to 1.5 mL Eppendorf tube containing 3 μL of 10 mM wact-469 (final concentration 60 μM), made up to final 1% DMSO and final volume of 500 μL with liquid NGM. The tubes were rocked gently in the dark on a nutator for 3 hours at RT. Worms were then washed twice with fresh M9, and paralyzed with 50 mM levamisole. Live worms were then mounted on a 3% agarose pad on a

glass slide and a coverslip was applied. The samples were viewed at 20X and 40X dry objectives with the Leica DMRA compound microscope. Wact-469 fluorescence was observed in the pharynx using the CFP (blue) channel and birefringence of wact-469 crystals were confirmed by polarizing filters.

## Nile red and Hoechst 33342 staining

Synchronized L1s of different genetic backgrounds were plated onto 10 cm plates at 6,500 L1s per plate, seeded with 1 mL OP50 bacteria and allowed to grow to young adults at room temperature (~ 3 days). Adult worms were washed off plates with M9, removing excess bacteria, and pelleted to ~20 worms/μL. 50 μL of the worm pelleted was added to 1.5mL Eppendorf tubes containing either 5μg/mL of Nile Red (Sigma-Aldrich) in 1% DMSO or 5 μg/mL of Hoechst 33342 (Cell Signaling Technology, USA) dissolved in ultrapure water, made to a final volume of 500 μL with liquid NGM. The tubes were then rocked gently in the dark on a nutator for 2 hours at RT. Worms were then washed twice with fresh M9, and paralyzed with 50 mM levamisole or 100 mM of sodium azide. Live worms were then mounted on a 3% agarose pad on a glass slide and a coverslip was applied. Worms were viewed at 20X and 40X objectives with a Leica DMRA compound microscope. Nile Red signal was captured using the Texas Red channel; Hoechst 33342 signal was captured using the A4/DAPI channel. All samples were analyzed using identical exposure times, magnifications and filters. Fluorescence was quantified using ImageJ.

## Mass spectrometric analyses of wactive accumulation

LCMS analyses were conducted as previously described [20]. Briefly, we synchronized *C. elegans* worms of the desired genotype as described above. Resulting L1s were grown on 10 cm plates, with 10,000 L1s per plate, and incubated at room temperature for 48 hours. Worms were then washed off the two plates with M9 buffer, collected into a 15 ml conical tube and washed three times. In parallel to this, we prepared a 'drug'-incubation buffer by first inoculating 400 ml of LB with 50 μl of an overnight culture of E. coli (HB101), grew it overnight, and centrifuged 50 ml of this fresh culture for 10 min at 2100 x g. We decanted the LB and rinsed the bacteria in ~50 ml of NGM buffer once and then resuspended the bacteria in 25 mls of NGM. We resuspended 20,000 worms in 1 ml of the NGM + HB101 solution in either 1% DMSO or 30 μM of wact-190 small molecule (to a final DMSO concentration of 1%) in 1.5 ml siliconized microcentrifuge tubes. We then incubated the tubes for 4 hours at 20°C on a nutator. Thereafter, we washed the worms five times with ice-cold M9 buffer, keeping the samples on ice as much as possible. We removed the M9 and flash froze the samples using liquid nitrogen and stored the worms at -80°C until the samples were ready to process by LCMS.

To perform an MTBE extraction of small molecules from the worm lysate, we first lyophilized the worm pellet and resuspended it in 200 μL of 0.1M NaCl, followed by sonication for 5 min at 100W in a Misonix cup sonicator. Samples were then spiked with 43 ng of internal standard (triamcinolone acetonide) and extracted twice with methyl tert-butyl ether (sample: MTBE, 1:5, v/v). Organic layers were pooled and evaporated to dryness under $N_2$. Extracts were re-suspended in 200 μL of HPLC-grade methanol and analyzed as described below.

The samples were analyzed by LC/MS/MS using a 6410 LC/MS/MS instrument (Agilent Technologies) with an ESI source in positive ion mode. Samples were separated on a Zorbax XDB-C18 column (4.6 x 50 mm, 3.5 μm) at 0.4 mL/min. The mobile phase consisted of HPLC-grade water (A) and methanol (B) both containing 5mM $NH_4Ac$. The following gradient was run: 0–1 min, 60% (B); 1–3.3 min, 60% to 100% (B); 3.3–7 min, 100% (B); stop time, 10 min; post-time, 5.5 min. MS parameters were as follows: nebulizer pressure 35 psi, drying

gas (nitrogen) 10 L/min, VCap 6000 V, Delta EMV 800V, column temperature 40°C and drying gas temperature 350°C for all compounds. For each of the following molecules, the following transitions were measured using multiple reaction monitoring (following the molecule's name, the retention-time (minutes), MRM (m/z), Fragmentor (voltage) and Collision Energy (voltage) are provided): Triamcinolone Acetonide, 5.7, 435➔415, 108, 5; wact-190, 7.7, 330➔107, 155, 22.

The extraction efficiency of wact-190 was 89% in MTBE. Our method was validated using external calibration curve with an internal standard of triamcinolone acetonide. The assay was linear between 10 uM and 50 uM (exhibiting non-linearity at 100uM). We estimated the LOD from the equation LOD = 3.3*Sy/S. By this measure the assay would have an LOD of 5 uM.

## Staining worms with GFP::NT-Lysenin protein

See Kamal *et al* (2019) [20] for the production of GFP::NT-Lysenin. Staining of animals were conducted as previously described [20]. Briefly synchronized adults were fixed in 1.5 ml Eppendorf tubes using the Modified Finney-Ruvkun protocol [76] except that all washes were done with tris-triton buffer (TTB) lacking triton-X-100; triton-X-100 was also replaced with 1X PBS in Buffers A and B in the final steps. The quality of fixation was tested by actin staining of fixed worms following 2hr incubation at RT with 0.328 nM phalloidin.

To stain worms with GFP::NT-Lysenin, 20 μL of fixed worms were suspended in 0.5 mg/mL of GFP::NT-Lysenin in the final volume of 100 μL of 1X PBS in a 0.6 mL tube. Fixed worms were incubated at 4°C for 4 hours in the dark with constant mixing. Worms were washed two times with 5.5 mL ice-cold 1X PBS. 10 μL of worms were mounted on a glass slide containing 3% agarose and a cover-slip was applied. Animals were imaged in the GFP channel using 20X magnification and identical exposure times for all strains. The experiment was repeated at least three times. Fluorescence intensity in the anterior pharynx was quantified using ImageJ software. For each worm, fluorescence intensity was determined by subtracting the background mean gray value (MGV) from MGV of the procorpus. The Student's t-test was performed on mean values of the different trials using GraphPad 6.0.

## Transmission Electron Microscopy (TEM) analyses

TEM analyses were conducted as previously described [20]. Briefly, approximately one thousand synchronized L1-stage worms were added to 60 mm plates containing 60 μM of wact-190 or 1% DMSO vehicle control seeded with OP50 bacteria. 24 hours later, worms were collected and prepared for transmission electron microscopy [77]. Live animals were loaded into a metal planchette in a slurry of *E. coli* and fast frozen under high pressure using a Balt-tec HM 010 freezing device, placed into 2% osmium tetroxide, 0.1% UAc, 2% dH2O in acetone at -90°C for 96 hrs using a Boeckler freeze substitution unit, slowly warmed (5°C per hr) to -30°C, held 16 hrs, then slowly warmed (5°C per hr) to 0°C, then rinsed in cold acetone at 0°C, and again rinsed several more times at room temp before infiltration into plastic resin over 3 days (HardPlus Embed812). Once fully infiltrated, the samples were cured at 60°C for 2 days and thin sectioned (mostly 70 nm) for views at high resolution using a Philips CM10 electron microscope with an 4X4 megapixel Morada side mount camera. Some samples were viewed in cross-section, and others in lengthwise section for comparison. The support films are made either from formvar (0.75%-1% formvar in 1,2-Dichloroethane solution) or a mixture of Pioloform:formvar (0.5% Pioloform +0.5% formvar in 1,2-Dichloroethane solution). The grids are coated with carbon ranging from 4–10 nm thickness. Wide slot grids (2000X1000um) from EMS or Ladd Research Inc are used. Control animals were exposed to DMSO without drug for 24 hrs.

## PGP-14 homology modeling

A *C. elegans* PGP-14 protein structure homology model was generated using the SWISS-MODEL pipeline [78]. The full-length amino acid sequence of PGP-14 was modeled using an experimental structure of human ABCB4 in the occluded conformation bound to phosphatidylcholine (PDB: 7NIV) as a template [64]. The structures were aligned and visualized using the PyMOL Molecular Graphics System (Version 2.5.0 Schrödinger, LLC).

## Statistics and graphs

Unless otherwise indicated, statistical differences were measured using a two-tailed Students T-test. Dot plots were generated using Prism 8 graphing software.

## Supporting information

**S1 Fig. Multiple Sequence Alignments with PGP-14. A.** A multiple sequence alignment with the indicated proteins using NCBI's Clustal Omega tool. All sequences were downloaded from Uniprot and are as follows: Human ABCB1(aka MDR1)(P08183), ABCB4(aka MDR3) (P21439), ABCB5 (Q2M3G0), ABCB11 (BSEP)(O95342), and *C. elegans* PGP-2 (G5EDF9), PGP-9 (G5EG58), PGP-12 (Q19733), PGP-13 (Q19734), PGP-14 (G5EG61). Black underline indicates the alpha-helixes that are the presumptive PGP-14 transmembrane domains as predicted by Alpha-fold. Red underline indicates the ABC nucleotide binding domain as predicted by the SMART protein database. Also highlighted are the non-sense (red highlight) and missense (blue highlight) mutations identified in the forward genetic screen. The blue boxes around the select residues of ABCB4 are those that interact with phosphatidylcholine as shown by cryo-EM [64]. The blue and green boxes around the select residues of ABCB11 are those that interact with bile acid as shown by cryo-EM [79]. **B.** Multiple sequence alignments of select regions of sequence of all *C. elegans* 14 PGPs aligned with each of three human ABCBs. The alignments show only the residues that have been implicated in substrate binding of the respective ABCB transporter. The ABCB1 (MDR1) alignment shows the residues (black boxes) important for interaction with taxol [32] and/or vincristine [33]. The ABCB4 alignment shows residues (black boxes) that interact with phosphatidylcholine [64]. The ABCB11 (BSEP) alignment shows residues (black boxes) important for interaction with bile salt [79]. The Uniprot entries used are as follows: PGP-1 (P34712), PGP-2 (G5EDF9), PGP-3 (P34713), PGP-4 (Q20331), PGP-5 (Q17645), PGP-6 (Q22656), PGP-7 (Q22655), PGP-8 (G5EG03), PGP-9 (G5EG58), PGP-10 (Q18824), PGP-11 (Q19015), PGP-12 (Q19733), PGP-13 (Q19734), PGP-14 (G5EG61), ABCB1(aka MDR1)(P08183), ABCB4(aka MDR3)(P21439), ABCB5 (Q2M3G0), and ABCB11 (BSEP)(O95342). In the alignment, red letters represent small and hydrophobic residues, blue letters represent acidic residues, magenta letters indicate basic residues, and green letters represent hydroxyl, sulfhydryl and amine residues. In each alignment, the last residue of PGP-14 is indicated to the right for reference. **C.** Cryo-EM structure of human ABCB4 (PDB ID 7NIV) [64] phosphatidylcholine (PC) binding pocket. ABCB4 is shown as ribbons (green), with key residues of the binding pocket labeled and shown as sticks. Bound PC is shown as sticks with an overlaid surface representation (pink). **D.** A *C. elegans* PGP-14 homology model of the ABCB4 PC binding pocket. This PGP-14 homology model is overlaid with the bound PC structure shown in (C). PGP-14 is shown as ribbons (blue). The PGP-14 residues aligning with those highlighted in (C) are labeled and shown as sticks. Key residues surrounding the choline moiety in the ABCB4 binding site (A231, W234, F345, H989) do not share identity between ABCB4 and PGP-14. This may impact recognition of PC and/or result in steric hindrance to PC binding. Further steric clashes in the binding site may occur as the result of bulkier side chains in PGP-14 than the corresponding ABCB4 residue

(ex. I736/F790).
(DOCX)

**S2 Fig. A Phylogenetic Tree Analysis of *C. elegans* and Human ABCB Family Members.** A phylogenetic tree of the indicated proteins constructed and displayed using MEGA11 [80] and ITOL [81] software. *C. elegans* proteins are highlighted in blue; human ABCB family proteins are highlighted in red. The pairwise blastp percentage identity (using NCBI protein blast tool) between PGP-14 and the four human ABCB family members is indicated next to the respective human protein names. The Uniprot entries used to create the phylogenetic tree and to make the pairwise blastp comparisons are the same described in S1 Fig.
(PDF)

**S3 Fig. Wact-190-Resistance is Specific to *pgp-14* Hypomorphic Mutations.** Top and middle sections: The behavior of 25 mutants harboring mutant genes of the ABC family, including 13 other *pgp* genes and the *pgp-15* pseudogene on solid or liquid media containing the indicated concentration of wact-190. Bottom section: Nine other missense mutations that were randomly generated in the million mutation project [28] that do not resist the lethality of wact-190, suggesting that the mutations fail to create a hypomorphic state.
(PDF)

**S4 Fig. PGP-14 Expression is Restricted to the Anterior Pharynx.** An RP3510 *trEx1010* [pPRHZ1138(*pgp-14p*::SMS-5::FLAG::mCherry); pPRHZ1144(*pgp-14p*::YFP-PGP-14)] adult worm stained with calcofluor white (CFW) highlighting that PGP-14 is expressed exclusively in the anterior pharynx. **A.** DIC image. **B.** The channel showing tagged-PGP-14 expression. **C.** The overlap in signal between tagged-PGP-14 and CFW. Orange arrows indicate the buccal cavity, red arrows indicate tagged-PGP-14 expression, and the yellow arrows highlight the pharynx terminal bulb grinder.
(PDF)

**S5 Fig. Additional Examples of the Localization of tagged-PGP-14 and SMS-5. A-F.** The head of an RP3510 *trEx1010*[pPRHZ1138(*pgp-14p*::SMS-5::FLAG::mCherry); pPRHZ1144 (*pgp-14p*::YFP-PGP-14)] animal before the L4 to adult molt (45 hours after synchronization at 20˚C). **G-L.** The head of an RP3510 just after the L4 to adult molt (the animal has one embryo in its uterus) (54 hours after synchronization). DIC, differential interference contrast; calcofluor white (CFW) and the functional tagged SMS-5 are shown in green; functional tagged PGP-14 is shown in fuchsia. Merge 1 is the merge of the PGP-14 and SMS-5 micrographs, and merge 2 is the merge of the PGP-14 and CFW micrographs. In all panels, orange arrows indicate the buccal cavity; the red arrows indicate an apical plane of the channels and marginal cells; the pink arrows indicate a basal lane of the marginal cells; the white arrows indicate the signal overlap between PGP-14 and SMS-5 that can be seen at the presumptive plasma membrane. The brightness and contrast of CFW before (B) and after (H) the molt are not manipulated (hence the 1X notation); the brightness and contrast of the tagged-PGP-14 signal before the molt (C) had to be increased by 50% to be clearly visible on the figure (hence the 1.5X notation); the brightness and contrast of the tagged-PGP-14 signal shortly after the molt (I) was not manipulated (hence the 1X notation); the lowly-expressed tagged-SMS-5 signal has been digitally increased to allow visibility (hence the >2X notation). **M.** Quantification of YFP:: PGP-14 expression in animals occurred over two trials with the indicated number of animals in each trial (each dot represents a single animal) during the inter-molt period and with animals that had clear evidence of molting (molt) (see methods for details). A Student's t-test was used to measure the significance of the difference.
(PDF)

**S1 Data. Tab A. The Whole-Genome Sequencing Data of *pgp-14* Mutants that were Recovered in Screens for Mutants that Resist wact-190.** The missense mutations are shown in columns L through AJ. The deletion alleles and rearrangements are shown in columns AL-AO. The data are summarized in Fig 3A in the main display items. **Tab B. The 2716 *C. elegans* Genes that Oscillate in their Expression.** The 2716 genes that were previously described to oscillate in their expression in a regular pattern at each larval stage [39] are shown. Column H shows the relative time at which the gene peaks in expression and column G shows the relative amplitude of the expression level. Genes whose expression is enriched during the L4-to-adult molt [15] are indicated in column J. Genes whose expression is enriched in the pharynx [16, 35] are indicated in column K. Genes that encode a signal peptide are shown column L. The data are summarized in Fig 5 in the main display items. **Tab C. The 508 Chembridge Inc Small Molecules Surveyed for Sensitivity in wild type and *pgp-14(ok2660)* and *sms-5 (ok2498)* Null Mutants.** Each strain was tested against each molecule four times in 96 well plate format. Each well is seeded with 20 synchronized L1 animals and scored 6 days later. A score of 9 (green) indicates the culture was overgrown (>>50 animals). A score of 0 indicates the culture had 11 or fewer animals (purple). The number of animals in wells with between 11 and 50 are indicated. The class of molecule (resistant, hypersensitive, etc) is shown in column E. These data are summarized in Fig 7E in the main display items. **Tab D. Predictions of whether the 48 Chembridge Inc Small Molecules (Wactives) that Form Crystals are Likely MDR1 (aka PGP, aka ABCB1) Substrates.** The 48 crystalizing wactives and their Chembridge identification numbers and SMILES structure are shown in columns A-D. The SwissADME [34] predictions for each molecule are shown in column M. **Tab E. The Predicted Hydrophobicity of 38 Crystalizing Molecules Compared to the 188 Hypersensitive Molecules.** The predicted hydrophobicity of the crystalizing and hypersensitive molecules is calculated in multiple ways as indicated. The means of the two groups as well as the statistical comparisons are also shown.
(DOCX)

## Acknowledgments

We thank Greg Fairn and Masashi Maekawa for reagents and guidance in characterizing sphingomyelin abundance and localization in the worm. We thank Don Moerman and Oliver Hobert for fosmids and fosmid recombineering tools; Chris Crocker for the artwork in Fig 1B; Andy Fraser and Michael Schertzberg for whole-genome sequence analysis. We thank Don Moerman, the *C. elegans* knock out consortium, the *C. elegans Genetics Centre* and Shohei Mitani for mutant strains.

## Author Contributions

**Conceptualization:** Muntasir Kamal, Peter J. Roy.

**Funding acquisition:** Duhyun Han, Carolyn L. Cummins, Peter J. Roy.

**Investigation:** Muntasir Kamal, Levon Tokmakjian, Jessica Knox, Duhyun Han, Houtan Moshiri, Lilia Magomedova, Ken CQ Nguyen, Hong Zheng, Andrew R. Burns, Brittany Cooke, Jessica Lacoste, May Yeo, David H. Hall.

**Methodology:** Muntasir Kamal, Jessica Knox, Duhyun Han, Houtan Moshiri, Lilia Magomedova.

**Project administration:** Peter J. Roy.

**Supervision:** Duhyun Han, David H. Hall, Carolyn L. Cummins, Peter J. Roy.

**Visualization:** Muntasir Kamal, Jessica Knox, David H. Hall, Peter J. Roy.

**Writing – original draft:** Peter J. Roy.

**Writing – review & editing:** Muntasir Kamal, Levon Tokmakjian, Jessica Knox, Duhyun Han, Houtan Moshiri, Lilia Magomedova, Ken CQ Nguyen, Hong Zheng, Andrew R. Burns, Brittany Cooke, Jessica Lacoste, May Yeo, Carolyn L. Cummins, Peter J. Roy.

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
