## [Decision Letter · Decision Letter 0]

28 Apr 2023

Dear Dr Roy,

Thank you very much for submitting your Research Article entitled 'The ABCB4 Homolog PGP-14 Establishes a Lipid Permeability Barrier within the C. elegans Pharyngeal Cuticle' to PLOS Genetics.

The manuscript was fully evaluated at the editorial level and by independent peer reviewers. The reviewers appreciated the attention to an important problem, but raised concerns about the current manuscript. Based on the reviews, we will not be able to accept this version of the manuscript, but we would be willing to review a much-revised version that addresses points raised below. We cannot, of course, promise publication at that time.

Should you decide to revise the manuscript for further consideration here, your revisions should address the specific points made by each reviewer. We will also require a detailed list of your responses to the review comments and a description of the changes you have made in the manuscript. Also, as academic editor I suggest that you 1) use the acknowledgement wording requested by the C. elegans Genetics Center, 2) correct nomenclature typos like capitalizing gene names the beginning of a sentence/non-italics in tables/etc, and 3) explicitly state when/if manual observations were made by observeds blinded to genotype/treatment. 

If you decide to revise the manuscript for further consideration at PLOS Genetics, please aim to resubmit within the next 60 days, unless it will take extra time to address the concerns of the reviewers, in which case we would appreciate an expected resubmission date by email to plosgenetics@plos.org.

We are sorry that we cannot be more positive about your manuscript at this stage. Please do not hesitate to contact us if you have any concerns or questions.

Yours sincerely,

Anne C. Hart

Academic Editor

PLOS Genetics

Gregory P. Copenhaver

Editor-in-Chief

PLOS Genetics

Reviewer's Responses to Questions

**Comments to the Authors:**

Reviewer #1: The manuscript ‘The ABCB4 Homolog PGP-14 Establishes a Lipid Permeability Barrier within the C. elegans Pharyngeal Cuticle’ by Kamal et al is an extension of their previous work (Kamal et al. 2019). In their previous work, they had shown that worm-active compounds form crystals or spheres and some of them cause lethality. Compounds such as wact-190 damage the plasma membrane. The previous study also reported a forward genetic screen to find genes that prevent lethality caused by wact. They had earlier identified mutations in sms-5 and pgp-14 to be involved in protection from crystals. This specific study is a further characterization of pgp-14 allele and complementation to show that the ABCB transporter is also protective. Pgp-14 is expressed in the same tissues and at the same time (in the context of molting) as sms-5. They work in the same pathway in protecting worms from toxic worm-active compounds.

This study lacks originality, it is a description of pgp-14 which works in the same pathway as sms-5 previously described. In fact, we are left in the same place where the previous study (Kamal et al. 2019) ended. The fact that lipid layer of the cuticle (as well as collagens) serve as a permeability barrier for genotoxic compounds has been demonstrated by various labs. Components of the lipid layer of the cuticle, sphingomyelin synthases and glycoproteins such as pgp-14, have long been ascribed to this function. In absence of findings that take the field forward, it is difficult to be enthusiastic about this study.

Other Comments:

Sms-5 was shown to have less sphingomyelin in their previous study. A similar phenotype has been shown for pgp-14 in a single worm image. There is no quantification or detergent sensitivity assay.

Does supplementation of sphingomyelin in the plates affect drug sensitivity or crystal formation.

Does supplementation of PC affect crystal formation? Is it dependent on both sms-5 and pgp-14?

The evidence that PGP-14 flops sphingolipid from one lipid layer to another is not provided. Indeed, the probe used, GFP_NT-lysenin does not report flopping at all, it just indicates binding to sphingolipids.

Reviewer #2: Most exposed epithelial surfaces are coated with an apical extracellular matrix (aECM) that confers barrier functions to protect against xenobiotics. This aECM barrier typically contains lipid components, though in many cases it is poorly understood what those lipids are, where they are located in the matrix, or how they get there. For example, in C. elegans, there are several different types of aECM that have barrier functions (eggshell, body cuticle, pharyngeal cuticle, intestinal glycocalyx), but none of these have well-defined lipid content or mechanisms of barrier assembly.

Here Kamal et al provide convincing evidence that the ABCB4-type transporter PGP-14 is needed to establish a proper xenobiotic barrier in the C. elegans pharynx cuticle, and that it acts in a common process with the sphingomyelin synthase SMS-5. Both proteins are expressed specifically in the pharynx, with peak expression during cuticle synthesis, and loss-of-function mutants have heightened sensitivity to a large number of xenobiotics, but reduced sensitivity to a smaller set of crystal-forming toxins. PGP-14 is related to a well-characterized phospholipid transporter, and reduced phospholipid content of the pharynx cuticle in mutants is inferred based on reduced staining with Nile Red and with a sphingomyelin sensor. Overall, the data appear rigorous and represent a significant step forward in understanding the biology of the pharyngeal cuticle - it will be of broad interest to cell and matrix biologists, especially those interested in lipid transporters. I am quite enthusiastic and have only minor suggestions to improve the manuscript:

1. There isn't any direct evidence for a specific affinity of PGP-14 for PC or for a change in PC levels or distribution in the mutants. Therefore I recommend toning down the suggestion that PC is the relevant phospholipid.

2. The MS starts out with a description of small molecule-induced toxic crystal formation and the isolation of pgp-14 mutants in a genetic screen for resistance to this. It's a beautiful example of the often unexpected insights that come from forward genetics, but it's also a rather circuitous route to the topic of aECM lipid content. It wasn't until very late in the MS (e.g. Discussion lines 366-371) that the authors clearly explained the likely connection between these topics (that hydrophobic lipids repel hydrophilic molecules but can also act as a sink to concentrate hydrophobic small molecules). It would be helpful if the Introduction better prepared the reader to make this connection!

3. As mentioned above, other worm tissues are coated by other types of aECM that also have lipid-based barriers - please state explicitly whether the tested crystal toxins do or do not concentrate in those other locations too. And if not, what is the hypothesis to explain this difference?

Minor comments:

Figure 4: The labelling of figure panels is fairly confusing here. Non-independent images (e.g. different channels from same image) should not have different panel letters.

Figure 5: revise title so that it doesn't imply that "genes" are secreted.

lines 207-208: The pgp-14 expression pattern is coincidental with established markers of the pharyngeal epithelium that INCLUDE..."

acknowledgements: "David Hall for Zeynep Altun" ?? DH is an author so shouldn't be thanked, not sure what this means

Reviewer #3: This manuscript, entitled “The ABCB4 Homolog PGP-14 Establishes a Lipid Permeability Barrier within the C. elegans Pharyngeal Cuticle”, fills important gaps in current knowledge about the role of ABC transporters in the nematode C. elegans.

These data are original, since Pgp14 deficiency is associated with resistance to small hydrophobic compounds, that forms crystals usually toxic in parental worms. Whereas the opposite is expected for conventional Pgp subtrate drugs, for which Pgp deficiency is generally associated with drug hypersensitivity, and overexpression with drug resistance.

The authors made several interesting observations, and conclude that Pgp-14 in C. elegans is the functional homolog of the mammalian transporter ABCB4. It is particularly intriguing that among the 14 Pgps in C. elegans, only Pgp-14 is essential for crystal formation and toxicity when exposed to the small hydrophobic toxic compounds wact. They identified several Pgp-14 null mutants which are resistant to the lethal effects of wact compounds, and there was no crystal formation in response to wact exposure in these mutants, as generally observed in the pharyngeal cuticle of parental worms. Interestingly, similar phenotype was also observed in worms deficient for sphingomyelinase SMS-5.

They provide a number of convincing evidences that Pgp14 may function coincidently with SMS-5 as a regulator of lipid membrane integrity. Pgp-14 co-regulates and locates close to SMS-5 in the pharynx cells. Pgp-14 deficiency leads to reduction of polar lipid as shown by Nil Red staining, and more specifically Pgp-14(0) have lower sphingomyelin abondance in the pharynx. In addition, lower penetration of Hoechst 33342 withing the pharynx cuticle of mutant worms is in favor of an altered lipid and Pgp transport function integrity. In agreement, CelPgp-14 protects the worms from several drug toxicity which is in favor of a drug efflux export function.

Points to be clarified

A major question remains: does Pgp-14 act indirectly on drug transport by changing lipid membrane composition? The data showing that Pgp-14 protects from several drug toxicity support a drug transport activity.

It remains unclear whether this is due to the specificity of tissue expression in the pharyngeal cuticle or to the characteristics of this specific transporter.

An additional experiment that would be informative is to rescue Pgp-14 deficient worms with a Pgp other than Pgp-14, chosen from the 13 others in C. elegans, and address it to the pharyngeal cells. This could be CelPgp-1 (Jin et al, 2012)... or any other Pgp known to be involved in drug transport. This will answer the question of the specific role of Pgp-14. It may be that all Pgp's can do the job, as long as they are expressed in the appropriate place.

The proposed model in the discussion seems to contradict the assertion that Pgp-14 is the functional homolog of ABCB4. While ABCB4 unambiguously transports PL, the authors propose that Pgp-14 does not. Rather SMS-5 produces PL in the lipid membrane, and the polar lipid enrichment enhances the activity of Pgp14. If this is the case, Pgp-14 will not be the exact functional homologue of ABCB4. Regarding the sequence comparisons, it is difficult to conclude of a sequence homology specific of PGP-14 and ABCB4. I will rephrase the title to moderate this aspect.

SFig1, comparing with other Pgp, the residues involved in PL binding on ABCB4 are not obviously present in Pgp-14.

1- The primary sequence of the protein does not allow necessarily to align hotspot residues. Alphafold 3D model structure would be more accurate.

2- This does not exclude the possible transport of other PL species. What do we know about PL composition in C. elegans? Is there a specific PL profile composition and species when compared to mammals?

Discussion line 385-386: the authors exclude the possibility that Pgp-14 transports wact crystals. In another way, Pgp14 may be able to export the free wact compounds that accumulate under the cuticle in wild type animals and form crystal that causes the toxicity. In Pgp14(0) the compound is not exported and may diffuse in the animal in a non-toxic form.

Can you discuss this point?

It would be of interest to discuss some of the published work repporting CelPgp14 expression and/or drug toxicity (Janssen et al, 2012; Guerrero et al 2021; Menez et al, 2016)

It is well known that mRNA levels do not necessarily correlate with protein levels. I would recommend tempering this aspect in the manuscript.

According to the table 4 in sup data file showing which wact- are Pgp substrates according to the Swiss ADME, would it be a rationale emerging from the wact compound structure that may explain their Pgp substrate specificity?

Fig3F shows the mass-spectrometry analyses which seem to correlate with the crystal content. Can you provide evidence that mass-spectrometry can measure free wact compounds and provide the yield of the extraction, and the limit of detection/quantification of the technique.

Minor points

It is unclear why in some experiments wact-90 is used and in others, wact-469.

Fig 3G represents fluorescence of wact compound. What is the difference when compared with the location of crystals showed by birenfringence in fig3C

Title of Figure 5. Change “secreted” by “expressed “

SFig1B: Please indicate which Pgp region is shown in this panel B

SFig2. The phylogenetic Tree Analysis of C. elegans and Human ABCB. Precise on which aa sequence the pairwise blast identity comparisons was performed. Indicate what the % corresponds to: identity or similarity.

Line 405: The gene products are RNA and not proteins: this should be clarified since transcript levels do not always reflect protein levels.

**Have all data underlying the figures and results presented in the manuscript been provided?**

Reviewer #1: Yes

Reviewer #2: Yes

Reviewer #3: Yes

PLOS authors have the option to publish the peer review history of their article (what does this mean?). If published, this will include your full peer review and any attached files.

Reviewer #1: No

Reviewer #2: No

Reviewer #3: No

---

## [Decision Letter · Decision Letter 1]

5 Oct 2023

Dear Dr Roy,

We are pleased to inform you that your manuscript entitled "PGP-14 Establishes a Polar Lipid Permeability Barrier within the C. elegans Pharyngeal Cuticle" has been editorially accepted for publication in PLOS Genetics. Congratulations!

Yours sincerely,

Anne C. Hart

Academic Editor

PLOS Genetics

Gregory P. Copenhaver

Editor-in-Chief

PLOS Genetics

Comments from the reviewers (if applicable):

Reviewer's Responses to Questions

**Comments to the Authors:**

Reviewer #2: The authors have satisfactorily addressed all of my prior concerns. I congratulate them on an interesting study!

Reviewer #3: no comments

**Have all data underlying the figures and results presented in the manuscript been provided?**

Reviewer #2: Yes

Reviewer #3: Yes

PLOS authors have the option to publish the peer review history of their article (what does this mean?). If published, this will include your full peer review and any attached files.

Reviewer #2: No

Reviewer #3: **Yes: **Anne Lespine

**Data Deposition**

http://datadryad.org/submit?journalID=pgenetics&manu=PGENETICS-D-23-00251R1

**Press Queries**

---

## [Editor Report · Acceptance letter]

26 Oct 2023

PGENETICS-D-23-00251R1 

PGP-14 Establishes a Polar Lipid Permeability Barrier within the C. elegans Pharyngeal Cuticle 

Dear Dr Roy, 

We are pleased to inform you that your manuscript entitled "PGP-14 Establishes a Polar Lipid Permeability Barrier within the C. elegans Pharyngeal Cuticle" has been formally accepted for publication in PLOS Genetics! Your manuscript is now with our production department and you will be notified of the publication date in due course.

With kind regards,

Anita Estes

PLOS Genetics

On behalf of:
